# Parameter-Masked Decoupled Optimization for Cross-Domain Class-Incremental Learning

Ziqi Gu [1]  Chunyan Xu [1] [*]  Yangguang Liu [1]  Wenxuan Fang [1]  Baotong Su [1]
Tong Zhang [1]  Dan Wang [2] [*]  Zhen Cui [3]

## Abstract

Cross-domain class-incremental learning (CD-CIL) requires models to continuously acquire new classes across shifting domains while retaining previously learned knowledge. Existing approaches often entangle what to update with how to update, resulting in unstable adaptation and severe forgetting under domain shifts. Inspired by the hippocampal learning mechanism that separates rapid adaptation from stable consolidation, we propose Parameter-Masked Decoupled Optimization (PMDO) that disentangles what knowledge is adapted from how learning proceeds in cross-domain class-incremental learning. We introduce a domain-aware knowledge decoupler that selectively adapts domain-relevant shared parameters, constraining incremental updates while preserving prior representations. To regulate how learning proceeds, we further design a stability-aware trajectory regulation that guides optimization along transferable and stable optimization trajectories, thereby reducing interference across domain transitions. PMDO enables effective cross-domain adaptation while mitigating catastrophic forgetting and maintaining long-term learnability. Extensive experiments across multiple benchmarks demonstrate the effectiveness of PMDO and its superiority over state-of-the-art methods.

## 1. Introduction

Recent advances in deep learning have driven substantial progress in computer vision (He et al., 2016; Hu et al., 2024),

further accelerated by large-scale foundation models (Wang et al., 2024; Hu et al., 2025) and extensive empirical validation. However, the standard fully supervised paradigm is increasingly impractical in dynamic settings: incorporating newly arriving data typically requires retraining (or heavily fine-tuning) on the combined historical and current datasets, leading to substantial computational and storage overhead.

Incremental learning (Shin et al., 2017; Li & Hoiem, 2017; Gu et al., 2025), also known as continual learning, addresses this tension by decomposing training into a sequence of tasks, each exposing the learner only to the newly observed classes. Many methods (Lopez-Paz & Ranzato, 2017; Gu et al., 2023b) have been developed to tackle continual learning, spanning knowledge-preserving strategies for structured representations such as knowledge graphs (Sun et al., 2023a; Li et al., 2024), self-supervised learning (Fini et al., 2022), and replay data (Rolnick et al., 2019; Zheng et al., 2025). Nevertheless, the dominant assumption remains that tasks are drawn from a single, homogeneous domain and arrive in batches; how to sustain continual learning when the domain itself shifts across tasks is still comparatively understudied.

In contrast, cross-domain class-incremental learning (CD-CIL)—the focus of this work—considers a sequence of heterogeneous domains, where a model must acquire new classes under substantial distribution shifts without access to data from previous domains. Recent studies have explored leveraging vision–language models for this setting. For instance, ZSCL (Zheng et al., 2023) integrates zero-shot generalization into CLIP to alleviate knowledge degradation, while MoE-Adapters (Yu et al., 2024) introduces task-specific components via a mixture-of-experts mechanism (Masoudnia & Ebrahimpour, 2014) to improve cross-domain adaptation. LEBA (Gu et al., 2026) mitigates forgetting by building bridging connections to capture relationships among adapters, and GIFT (Wu et al., 2025) further reduces forgetting through diffusion-synthesized image–text replay and distillation. Despite their effectiveness, most existing approaches sustain continual adaptation by progressively stacking task-specific adapters or auxiliary components. This design inevitably increases model complexity and computational overhead, complicates optimization as

---

[1]School of Computer Science and Engineering, Nanjing University of Science and Technology, Nanjing, China. [2]Beijing Institute of Spacecraft System Engineering, Beijing, China. [3]School of Artificial Intelligence, Beijing Normal University, Beijing, China.. [*]Corresponding authors: Chunyan Xu <cyx@njust.edu.cn>, Dan Wang <wangdan_ict_hit@163.com>.

*Proceedings of the 43$^{rd}$ International Conference on Machine Learning*, Seoul, South Korea. PMLR 306, 2026. Copyright 2026 by the author(s).

the task sequence grows, and can even aggravate forgetting due to the accumulation of domain-specific parameters.

Fundamentally, the central difficulty of CD-CIL is to sustain a viable stability–plasticity trade-off under domain shifts without maintaining a separate adapter for each domain. The incremental learner must rely on a shared parameter space to preserve previously acquired representations while absorbing new classes as the data distribution changes substantially across domains. Prior studies (Dohare et al., 2024; Shi et al., 2025) have indicated that aggressively updating a large fraction of shared parameters on a new domain can substantially exacerbate forgetting and hinder subsequent learning; moreover, as incremental training proceeds, many shared parameters may gradually become "inactive" and less responsive to new knowledge, reducing plasticity and shrinking the effective update space. These observations highlight two remaining challenges: (i) how to adaptively identify a domain-relevant subset of shared parameters that truly needs to be updated; (ii) how can we regulate parameter updates via directional constraints or guidance, ensuring the optimization trajectory is transferable and stable while minimizing disruption to previously learned knowledge?

To address the above challenges, we propose Parameter-Masked Decoupled Optimization (PMDO), a unified framework for cross-domain class-incremental learning inspired by the hippocampal principle (Manjón et al., 2022; Ballard et al., 2019) of separating rapid adaptation from stable consolidation. PMDO explicitly disentangles what to update from how to update under domain shifts: when a new domain arrives, it adaptively identifies a domain-relevant subset of shared parameters that genuinely requires updating and restricts optimization to this necessary scope, thereby improving adaptation while keeping previously learned representations as stable as possible. Concretely, we introduce a domain-aware knowledge decoupler that constructs a lightweight domain-relevance mask to selectively activate learnable parameters, avoiding wholesale fine-tuning of shared weights and preserving prior representations. However, selective updates alone are insufficient at domain boundaries, where abrupt shifts in data distributions and training objectives can steer gradients toward directions that conflict with previously learned knowledge, leading to unstable optimization and accelerated forgetting. To this end, we further design a stability-aware trajectory regulation module that constrains update directions, guiding optimization along transferable and stable trajectories so that the model can absorb new-domain knowledge without unnecessary disruption to prior representations and decision boundaries.

In summary, our primary contributions are four-folds: i) propose a parameter-masked decoupled optimization framework for CD-CIL, which disentangles update scope from update trajectory to enable efficient cross-domain adapta-

tion while mitigating catastrophic forgetting; ii) design a domain-aware parameter decoupler that adaptively selects a domain-relevant shared parameter subset for updating, restricting incremental adaptation to a necessary scope; iii) introduce a stability-aware trajectory regulator that steers the selected updates along transferable trajectories, reducing interference with previously learned representations under domain shifts; iv) report new state-of-the-art performance across two CD-CIL task settings on multiple benchmarks.

## 2. Related work

**Cross-domain Class Incremental Learning:** In contrast to conventional incremental learning, which has centered on knowledge acquisition within a single domain, cross-domain class-incremental learning has required sequentially learning from multiple heterogeneous domains. Under this setting, the incremental model (e.g., a vision–language model) has needed not only to learn new tasks over time and alleviate catastrophic forgetting, but also to transfer knowledge effectively across diverse domains. Along this line, ZSCL (Zheng et al., 2023) has employed parameter regularization for incremental learning in large-scale models. MoE-Adapters (Yu et al., 2024) have enhanced CLIP by integrating task-specific components, thereby improving adaptability. Xu et al. (Xu et al., 2024b) have proposed RAIL, a recursive ridge regression approach that has enabled non-forgetting incremental learning by projecting features into a higher-dimensional space. LEBA (Gu et al., 2026) has mitigated catastrophic forgetting by establishing bridging connections that explicitly capture relationships among adapters. GIFT (Wu et al., 2025) has combated forgetting in VLMs through diffusion-synthesized image–text replay and distillation.

**Class Incremental Learning:** Previous work on incremental learning has focused on designing diverse architectures (Chen & Zhou, 2025; Gu et al., 2023a), broadly including memory-based, regularization-based, and dynamic models. Memory-based methods have preserved historical knowledge by maintaining a memory bank that has been periodically retrieved and updated during incremental training (Hersche et al., 2022; Zheng et al., 2025; Lavda et al., 2018; Li & Zhou, 2025). Regularization-based methods have introduced explicit constraints on model weights to balance prior and incoming tasks (Aljundi et al., 2018; Kirkpatrick et al., 2017; Zhang et al., 2025), or have imposed regularization at the data level (Hou et al., 2019; Li & Hoiem, 2017). Dynamic methods have addressed incremental learning by progressively expanding the base model with additional parameters—such as new neurons, branches, or task-specific heads—across stages (Hu et al., 2023; Ye & Bors, 2023; Yan et al., 2021; Li et al., 2025).

**Vision-Language Models:** Recently, research on vi-

sion–language models (VLMs) has rapidly evolved from foundational architectural alignment to capability enhancement and domain generalization. Early efforts (Fang et al., 2025; Fei et al., 2024; Shinoda et al., 2025) have focused on efficient instruction tuning to align large language models with visual encoders, enabling cost-effective multimodal understanding and generation. VLMs have been increasingly applied to complex real-world scenarios, including embodied intelligence and spatial reasoning, where models (Cachet et al., 2024; Zhang et al., 2024) have been shown to follow environment-grounded language instructions in a zero-shot manner. Comprehensive evaluation has become essential, leading to the development of large-scale benchmarks such as LVLM-eHub (Xu et al., 2024a) for standardized, multi-dimensional assessment. These evaluations have also exposed vulnerabilities (Liu et al., 2025), motivating growing interest in the security and robustness of VLMs against adversarial attacks. In addition, VLMs have also been widely adopted in incremental learning. GMM (Cao et al., 2024) and MGMM (Cao et al., 2025) approaches have demonstrated that VLMs can support effective incremental learning, and Xue et al. (Xue et al., 2025) have applied CLIP to few-shot class-incremental learning task.

## 3. The Proposed Method

### 3.1. Problem Formulation

Cross-domain class incremental learning (CD-CIL) learns classes sequentially from a stream of labeled domains, where data from earlier domains becomes unavailable in later stages. It evaluates both the model's ability to adapt to newly arriving classes and its robustness to catastrophic forgetting. Formally, given a sequence of $T$ task domains $\{\mathcal{S}^t\}_{t=1}^T$. At the $t$-th session, we aim to learn an incremental model $\Theta^t$ using the data from the current task domain $\mathcal{S}^t$. Each domain $\mathcal{S}^t$ is characterized by a labeled dataset $\mathcal{D}^t$ and a caption set $\mathcal{C}^t$, i.e., $\mathcal{S}^t := (\mathcal{D}^t, \mathcal{C}^t)$. The dataset is composed of input–label pairs, written as $\mathcal{D}^t := \{(x_i^t, y_i^t)\}_{i=1}^{N_t}$, where $(x_i^t, y_i^t)$ denotes the $i$-th sample and $N_t$ is the number of training instances in $\mathcal{S}^t$. The caption set $\mathcal{C}^t := \{c_j^t\}_{j=1}^{M_t}$ encodes the semantic descriptors of domain $\mathcal{S}^t$ (e.g., class-related semantics associated with labels $y_i^t$), where $M_t$ denotes the number of unique class names. In this incremental paradigm, task domains are usually disjoint in terms of class labels, i.e., for any two domains $\mathcal{S}^i$ and $\mathcal{S}^j$, we have $\mathcal{C}^i \cap \mathcal{C}^j = \varnothing$ when $i \neq j$. A standard baseline for CD-CIL is to fine-tune the model from the previous step on the current domain, i.e., $\Theta^t \leftarrow \Theta^{t-1} - \frac{\partial \zeta(\mathcal{S}^t)}{\partial \Theta}$, where $\zeta(\cdot)$ denotes a supervised objective (e.g., cross-entropy).

Nevertheless, striking an effective balance between rapid adaptation to newly arriving domains and mitigating catastrophic forgetting remains challenging: many existing methods (Gu et al., 2026; Yu et al., 2024) rely on progressively

---

**Algorithm 1** PMDO Training Procedure

**Input:** Task sequence $\mathcal{S}^t = \{(\mathcal{D}^t, \mathcal{C}^t) \mid t = 1, \ldots, T\}$; incremental model $\Theta^{t=1}$; parameter decoupler $\Gamma$; trajectory regulator $\pi$

**Output:** Incremental model $\Theta^T$; parameter decoupler $\Gamma$; trajectory regulator $\pi$;

1: Initialize model $\Theta^{t=1}$, regulator $\pi$, and decoupler $\Gamma$
2: **for** $t = 1$ to $T$ **do**
3:    # Supervised training on the current task
4:    Train $\Theta^t$ on $\mathcal{S}^t = (\mathcal{D}^t, \mathcal{C}^t)$
5:    **if** $t > 1$ **then**
6:      # Feature extraction
7:      Extract visual features $\mathbf{v}_i$ and textual features $\mathbf{z}_i$ as inputs
8:      # Domain-Aware Knowledge Decoupler
9:      Generate domain-relevance scores $\eta^{t,l}$ by decoupler $\Gamma$
10:     Decouple an learnable parameter subset $W^l$ from $\Theta^t$ via decoupler $\Gamma$
11:     # Stability-Aware Trajectory Regulator
12:     Construct a LoRA subspace $\Delta W$ and orthonormal bases $U_k$ and $V_k$
13:     Build an optimization trajectory regulator $\pi$
14:     Compute routing scores for each orthonormal bases and obtain confidence weights $\mathbf{q}_{i,k}^t$
15:     Align parameter update trajectory via regulator $\pi$
16:     # Joint Optimization
17:     Compute the total loss $\zeta_{\text{total}}$
18:     Update $\Theta^t$, $\pi$, and $\Gamma$ by minimizing $\zeta_{\text{total}}$ (Eqn. 13)
19:    **end if**
20:    # Update the incremental model for the next domain
21:    $\Theta^{t+1} \leftarrow \Theta^t$
22: **end for**

---

stacking task-specific adapters, which incurs steadily growing model complexity and amplifies both computational and optimization burdens. In contrast, we regulate cross-domain continual adaptation by explicitly decoupling what to update from how to update, and formulate CD-CIL as:

$$\arg \min_{\Theta^{t-1}, \Gamma, \pi} \underbrace{\zeta_S(\mathcal{S}^t; \Theta^{t-1})}_{\text{supervised optimization}} + \underbrace{\zeta_{\mathcal{D}}(g(x_i^t, c_i^t; \Gamma), h(x_i^t, c_i^t; \pi); \Theta^{t-1})}_{\text{domain-aware optimization}},$$

(1)

where the function $g(\cdot)$ performs parameter decoupling on the learnable components of the incremental model $\Theta$ by applying a decoupling operator $\Gamma$ to quantify domain-specific parameter relevance, thereby selecting the subset that remains plastic for optimization. Building on the decoupled parameter space, $h(\cdot)$ regulates how the selected parameters are updated by imposing directional guidance through the operator $\pi$, which steers optimization toward transferable and stable trajectories and suppresses cross-domain interference. This design is consistent with human learning, where new knowledge is acquired via context-aware separation and directionally regulated adaptation, rather than indiscriminate modification of existing representations.

### 3.2. Overview

Building on the formulation in Eqn. (1), our Parameter-Masked Decoupled Optimization (PMDO) framework com-

prises two tightly coupled modules: (i) a learnable update-scope decoupling mechanism that determines what to update by selecting a domain-relevant subset of shared parameters, and (ii) a trajectory regulation mechanism that determines how to update by constraining the optimization directions under domain shifts. Accordingly, we instantiate these two modules as the Domain-Aware Knowledge Decoupler (DKD) in Section 3.3 and the Stability-Aware Trajectory Regulator (STR) in Section 3.4. In DKD, rather than globally optimizing all parameters $\Theta^t$ using only the supervised objective $\zeta_S$, we introduce a decoupling operator $\Gamma$ that leverages a learnable parameters $\theta_p$ together with soft masks $\tilde{\mathbf{M}}$ to carve out an updatable subset $\Theta_m^t \subseteq \Theta^t$ and to explicitly quantify each parameter's relevance to the current domain. Guided by these relevance estimates, DKD selectively keeps domain-critical parameters plastic for adaptation while shielding the remaining parameters from domain-irrelevant perturbations, thereby preserving prior representations. In STR, we capture domain-specific optimization biases and inject them as trajectory regulation signals $\pi$, guiding updates of $\Theta_m^t$ toward transferable and stable directions. This regulation mitigates representation drift and cross-domain interference that often result from unconstrained fine-tuning at domain boundaries. Together, DKD and STR jointly constrain incremental learning from both the update scope and the update trajectory, enabling effective cross-domain adaptation while improving robustness to catastrophic forgetting. The overall PMDO training procedure is summarized in Algorithm 1.

### 3.3. Domain-Aware Knowledge Decoupler

In cross-domain class-incremental learning, shared parameters are crucial for knowledge transfer across domains; however, naïvely updating them on each incoming domain allows domain-shifted gradients to overwrite previously acquired representations, leading to severe forgetting and unstable optimization. Inspired by hippocampal mechanisms (Manjón et al., 2022): rather than indiscriminately modifying all neural pathways, it selectively adjusts experience-relevant circuits to acquire new information while preserving consolidated knowledge. Motivated by this principle, we argue that adaptation to a new domain should avoid global fine-tuning of shared parameters and instead confine updates to a domain-relevant subspace. To this end, we propose a Domain-Aware Knowledge Decoupler that selects a plastic subset of shared parameters for incremental optimization, thereby reducing domain-irrelevant perturbations and improving stability under domain shifts.

Given an input sample consisting of an image–caption pair $(x_i^t, c_i^t)$, we extract cross-modal semantic features using frozen encoders. The image and text are mapped into a shared semantic space, producing modality-specific embed-dings that jointly characterize the current domain context:

$$\mathbf{v}_i^t = \text{Enc}_{\text{img}}(x_i^t), \qquad \mathbf{z}_i^t = \text{Enc}_{\text{txt}}(c_i^t), \qquad (2)$$

where $\text{Enc}_{\text{img}}$ and $\text{Enc}_{\text{txt}}$ are image and text encoders, respectively. Based on concatenated features $[\mathbf{v}_i^t; \mathbf{z}_i^t]$, we infer a domain-dependent relevance feature through a lightweight neural network $\varepsilon(\cdot; \theta_p)$, which can be formulated as:

$$\eta^{t,l} = \varepsilon([\mathbf{v}_i^t, \mathbf{z}_i^t]; \theta_p). \qquad (3)$$

For each $l$-th layer with $c_l$ neurons, the neural network $\theta_p$ outputs a relevance score $\eta^{t,l} \in \mathbb{R}^{c_l}$, indicating how each neuron should be involved in adapting to the current domain.

To make the update scope explicit and learnable, we introduce a structural soft mask that is optimized jointly with the incremental model. Specifically, we maintain a learnable mask matrix $\tilde{\mathbf{M}}^{(l)} \in \mathbb{R}^{c_l}$, whose sigmoid-normalized form $\tilde{\mathbf{M}}^{(l)} \in (0,1)^{c_l}$ encodes a persistent update preference accumulated across incremental session, formally:

$$\mathbf{R}^{(l)}(x_i^t) = \text{Top-}k(\eta^{t,l}) \odot \tilde{\mathbf{M}}^{(l)}, \quad \mathbf{R}^{(l)}(x_i^t) \in [0,1]^{c_l}, \quad (4)$$

where $\mathbf{R}^{(l)}(x_i^t) \in [0,1]^{c_l}$ is a sample-specific soft mask. Specifically, we fuse the learned structural prior and sample-conditioned relevance scores through element-wise modulation, followed by a Top-$k$ operation that activates only the most relevant neurons. In this way, adaptation is confined to a compact domain-specific subspace, with $k$ controlling the update scope for the current domai The resulting mask $\mathbf{R}^{(l)}(x_i^t)$ gates gradient propagation at the neuron level, enabling selective parameter updates. Updates are applied only to the selected parameters, while the remaining ones are kept unchanged. Accordingly, we enforce a mask-based update rule, formally:

$$W^{(l)} \leftarrow \mathbf{R}^{(l)}(x_i^t) \odot W^{(l)}, \qquad (5)$$

where $W^{(l)}$ denotes the learnable parameters at $l$-th layer and $\mathbf{R}^{(l)}(x_i^t) \in [0,1]^{c_l}$ is the update mask. This mechanism confines adaptation to a necessary subset of parameters, thereby reducing unnecessary perturbations and cross-domain interference.

Restricting updates to a compact subset is essential for reducing cross-domain interference, yet overly conservative gating can collapse mask activations toward zero and hinder adaptation. To prevent this degenerate behavior, we introduce a budget regularizer:

$$\zeta_{\text{DKD}} = \left(\frac{1}{c_l}\sum_{j=1}^{c_l}\mathbf{R}_j^{(l)} - \rho\right)^2, \qquad (6)$$

which softly aligns the mean activation of the $l$-th mask with a target ratio $\rho$, i.e., the expected fraction of parameters permitted to participate in the update. This constraint avoids

gate collapse while retaining selective updates and sufficient learning capacity. With this objective, the decoupler separates the learnable parameters in the incremental model, enabling adaptation to new domains while minimizing unnecessary perturbations to learned representations.

### 3.4. Stability-Aware Trajectory Regulator

While the parameter decoupler constrains the update scope by activating only a compact subset of parameters, it does not specify how these parameters should be updated. In cross-domain incremental learning, gradients computed from the new domain can still be directionally inconsistent, leading to unstable optimization even within the selected subset. To address this issue, we introduce a stability-aware trajectory regulator that regulates the update trajectories, steering learning toward more transferable and stable directions and thereby suppressing cross-domain interference.

We focus on an incremental model weight $W \in \mathbb{R}^{d_v \times d_h}$. During incremental training, we take the pretrained weight $W_0$ and introduce a low-rank adaptation:

$$W = W_0 + \Delta W, \qquad \Delta W = AB^\top, \qquad (7)$$

where $A \in \mathbb{R}^{d_v \times r}$, $B \in \mathbb{R}^{d_h \times r}$, and $r \ll \min(d_v, d_h)$. Hence, all adaptation is governed by the updates of $(A, B)$. To explicitly regulate the optimization directions of LoRA (Hu et al., 2022) factors, we construct a bank of $K$ direction subspace experts. Each expert $(k)$ is represented by two orthonormal bases: $U_k \in \mathbb{R}^{d_v \times m}$ and $V_k \in \mathbb{R}^{d_h \times m}$, where $m \ll d_v, d_h$. where columns of $U_k$ and $V_k$ are mutually orthonormal. These bases define two projection operators: $P_k^{\text{in}} = U_k U_k^\top$ and $P_k^{\text{out}} = V_k V_k^\top$. Intuitively, $P_k^{\text{in}}$ constrains allowable update directions for $A$ (input-side), while $P_k^{\text{out}}$ constrains those for $B$ (output-side).

Different domains/classes may favor distinct adaptation directions. To capture this variability, we employ a lightweight router $\pi$ to produce sample-conditioned expert weights. Concretely, we fuse the cross-modal features to form a routing representation:

$$\mathbf{s}_i^v = f_v(\mathbf{v}_i^t; \theta_v), \qquad \mathbf{s}_i^t = f_t(\mathbf{z}_i^t; \theta_t), \qquad (8)$$

where $\mathbf{v}_i^t$ and $\mathbf{z}_i^t$ denote the visual and textual features extracted from the $i$-th sample $(x_i^t, c_i^t)$, respectively. $\mathbf{s}_i^v$ and $\mathbf{s}_i^t$ indicate the projection subspaces for the visual and textual features, respectively. The operator $\pi(\cdot; \theta_v, \theta_t)$ is implemented as a lightweight projection network with learnable parameters $\theta_v$ and $\theta_t$, which are used to parameterize the visual and textual projection functions $f_v(\cdot; \theta_v)$ and $f_t(\cdot; \theta_t)$, respectively. It serves as an optimization trajectory regulator that injects domain-dependent biases to steer the incremental model toward transferable and stable update trajectories. We compute the routing weights as:

$$\mathbf{q}_i^t = \text{softmax}(\text{MLP}[\mathbf{s}_i^t, \mathbf{s}_i^v]), \qquad \mathbf{q}_i^t \in \mathbb{R}^K, \qquad (9)$$

where $\mathbf{q}_{i,k}^t$ indicates the dependence of sample $x_i^t$ on the $k$-th direction subspace. Rather than selecting discrete experts, we form soft sample-specific projectors by convex mixing:

$$\bar{P}_i^{\text{in}} = \sum_{k=1}^K \mathbf{q}_{i,k}^t P_k^{\text{in}}, \qquad \bar{P}_i^{\text{out}} = \sum_{k=1}^K \mathbf{q}_{i,k}^t P_k^{\text{out}}. \qquad (10)$$

Let $\zeta_{total}$ denote the incremental training objective. The unconstrained gradients on LoRA factors are $\nabla_A \zeta$ and $\nabla_B \zeta$. We steer optimization by projecting gradients onto the routed subspace mixture:

$$\widetilde{\nabla}_A \zeta = \bar{P}_i^{\text{in}} \nabla_A \zeta, \qquad \widetilde{\nabla}_B \zeta = \bar{P}_i^{\text{out}} \nabla_B \zeta. \qquad (11)$$

This mechanism enforces that parameter changes are directionally confined: new-domain gradients are allowed to modify $(A, B)$ only along routed subspace directions, suppressing harmful components that would overwrite prior representations.

While the balancing constraint promotes diverse expert usage at the batch level, the routing distribution can remain diffuse at the sample level, with each sample assigning comparable weights to many subspaces. We therefore introduce a sharpness regularizer that encourages each sample to concentrate its probability mass on a small subset of experts. Given router outputs $\mathbf{q}_i^t \in \mathbb{R}^K$ with $\sum_{k=1}^K \mathbf{q}_{i,k}^t = 1$, we minimize the entropy of $\mathbf{q}_i^t$:

$$\zeta_{STR} = \frac{1}{K} \sum_{i=1}^K H(\mathbf{q}_i^t) = -\frac{1}{K} \sum_{i=1}^K q_{i,k}^t \log\left(\mathbf{p}_{i,k}^t + \epsilon\right), \qquad (12)$$

where $\epsilon$ is a small constant for numerical stability. By projecting LoRA gradients onto sample-adaptive direction subspaces, the regulator steers updates toward domain-consistent directions and filters out components that would otherwise corrupt previously learned representations. This design promotes stable, discriminative, and interpretable update patterns, facilitating cross-domain incremental adaptation while alleviating catastrophic forgetting.

### 3.5. Optimizing the PMDO

To optimize PMDO for cross-domain class incremental learning, we adopt a unified objective that directly addresses two primary failure modes under domain shifts: (i) indiscriminate updates to shared parameters that overwrite previously learned representations, and (ii) directionally inconsistent updates induced by new-domain data that destabilize optimization. Specifically, we minimize as:

$$\begin{aligned} \zeta_{\text{total}} = &\zeta_{\text{CE}}(X^t, Y^t; \Theta^t) \\ &+ \gamma \zeta_{\text{DKD}}(X^t, C^t; \Gamma) + \alpha \zeta_{\text{STR}}(X^t, C^t; \pi), \end{aligned} \qquad (13)$$

where $\zeta_{\text{CE}}$ denotes the cross-entropy loss on the current domain. The term $\zeta_{\text{DKD}}$ instantiates our domain-aware

knowledge decoupling by adaptively restricting updates to a domain-relevant subset of shared parameters, thereby reducing unnecessary perturbations to consolidated representations. Complementarily, $\zeta_{\text{STR}}$ implements stability-aware trajectory regulation by controlling how the selected parameters are updated, encouraging transferable and stable optimization directions at domain transitions and suppressing destabilizing drift. The balancing factors $\gamma$ and $\alpha$ trade off plasticity and stability, enabling reliable cross-domain incremental adaptation with reduced catastrophic forgetting.

# 4. Experiment

## 4.1. Experimental Setting

**Datasets:** We evaluate PMDO under the cross-domain class incremental learning setting, where tasks are drawn from heterogeneous visual domains and each domain requires distinct domain-specific knowledge for strong performance. Following the CD-CIL benchmark protocol, the stream consists of 9 domain tasks with 1,281 classes in total, including Aircraft (Maji et al., 2013), Caltech101 (Fei-Fei et al., 2004), DTD (Cimpoi et al., 2014), Flowers (Nilsback & Zisserman, 2008), Food (Bossard et al., 2014), Imagenet-R (Hendrycks et al., 2021), OxfordPet (Parkhi et al., 2012), StanfordCars (Krause et al., 2013), and SUN397 (Xiao et al., 2010). We consider two task orders. Order-I follows the alphabetical order: Aircraft → Caltech101 → DTD → Flowers → Food → Imagenet-R → OxfordPet → StanfordCars → SUN397. Order-II uses a random permutation: Food → SUN397 → Caltech101 → StanfordCars → Imagenet-R → Flowers → OxfordPet → DTD → Aircraft. Unless otherwise specified, we report results under Order-I.

**Implementation Details:** We follow the GMM (Cao et al., 2024) baseline by adopting a generative-model–based backbone. Specifically, we follow BLIP2 (Li et al., 2023) and use the EVA-CLIP (Sun et al., 2023b) pretrained ViT-g/14 together with the BLIP2-pretrained Q-Former. We additionally initialize the projection layer with the pretrained MiniGPT-4 checkpoint. The parameter decoupler $\Gamma$ consists of a lightweight one-layer MLP and a learnable soft mask. The trajectory regulator $\pi$ is implemented as an expert network composed of LoRA and a two-layer MLP. We set the balancing factors to $\gamma=0.5$ and $\alpha=0.1$, and use $\rho=0.2$, i.e., updating (20%) of the shared parameters at each layer. We adopt orthonormal bases as experts and set the total number of experts to ($K=36$), with 18 for the visual branch and 18 for the textual branch. We combine experts using top-2 gating scores. We optimize all models using AdamW (Loshchilov & Hutter, 2017) and apply label smoothing (Müller et al., 2019) to strengthen the baselines. For evaluating the model's performance, we use "Avg" metrics. The "Avg" metric is calculated as the mean of the Top-1 accuracy in all sessions as $\sum_{t=1}^{T} \text{Acc}_t/T$. For the

CD-CIL benchmark, we use a batch size of 64 and select the learning rate from $\{10^{-3}, 10^{-4}, 10^{-5}\}$. All experiments are implemented in PyTorch and run on NVIDIA GeForce RTX 4090 GPUs. We compare PMDO with representative baselines, including Continual-FT, LwF (Li & Hoiem, 2017), iCaRL (Rebuffi et al., 2017), WiSE-FT (Ding et al., 2022), ZSCL (Zheng et al., 2023), MoE-Adapters (Yu et al., 2024), and state-of-the-art methods such as LEBA (Gu et al., 2026), GIFT (Wu et al., 2025), and GMM (Cao et al., 2024).

## 4.2. Comparison with State-of-the-art Methods

Table 1 and Table 2 summarize the "Avg" results of all evaluated methods across domains under two CD-CIL benchmarks. While existing approaches can partially alleviate forgetting in cross-domain class incremental learning, many of them rely on progressively stacking domain-specific components to sustain adaptation. Such designs inevitably increase model complexity and training overhead, and as the domain sequence grows, optimization becomes more susceptible to gradient conflicts induced by domain shifts, which undermines the stability–plasticity trade-off and leads to aggravated forgetting.

Our PMDO achieves stronger "Avg" performance on most tasks in both benchmarks and consistently surpasses the strongest baseline GMM in terms of overall average, indicating more robust cross-domain adaptation and long-term learnability. These gains align with our core "what/how" decoupling principle. In the shared parameter space, PMDO first employs a domain-aware knowledge decoupler to adaptively identify the subset of shared parameters that is truly relevant to the incoming domain and genuinely needs to be updated, and it restricts incremental learning to this necessary scope. This prevents wholesale perturbations of shared weights and better preserves prior discriminative representations and decision boundaries. Meanwhile, selective updates alone can still suffer from directionally inconsistent gradients and unstable optimization at domain boundaries. To this end, PMDO further introduces a stability-aware trajectory regulation module to constrain and guide update directions, steering optimization along more transferable and stable trajectories. As a result, the model can absorb new-domain knowledge while minimizing disruption to previously learned knowledge, thereby mitigating catastrophic forgetting. Overall, the two modules act in a complementary manner by addressing what to update and how to update, enabling PMDO to deliver more stable preference improvements and a more reliable stability–plasticity balance in cross-domain class-incremental learning.

## 4.3. Ablation Study

This section focuses on analyzing the effectiveness of the proposed PMDO method. All experiments are conducted in a cross-domain class incremental learning setting, with

*Table 1.* Comparison with state-of-the-art methods on cross-domain class-incremental learning (Order-I).

| Method | Aircraft | Caltech101 | DTD | Flowers | Food | ImageNet-R | OxfordPet | Cars | SUN397 | *Average* |
|---|---|---|---|---|---|---|---|---|---|---|
| Continual-FT | 36.5 | 88.2 | 70.3 | 85.4 | 84.9 | 72.4 | 80.9 | 80.6 | 76.0 | 75.1 |
| LwF | 39.2 | 94.3 | 72.6 | 90.1 | 88.2 | 77.8 | 84.4 | 85.2 | 79.4 | 79.0 |
| iCaRL | 41.4 | 92.0 | 74.9 | 87.6 | 90.7 | 75.6 | 86.8 | 82.6 | 81.4 | 79.3 |
| WiSE-FT | 40.1 | 93.1 | 73.8 | 88.9 | 89.5 | 76.9 | 85.7 | 84.0 | 80.6 | 79.1 |
| ZSCL | 40.6 | 92.5 | 74.5 | 88.1 | 90.1 | 76.3 | 86.4 | 83.4 | 81.3 | 79.2 |
| MoE-Adapters | 51.1 | 92.1 | 73.0 | 85.9 | 83.1 | 74.1 | 83.1 | 83.5 | 74.9 | 77.9 |
| LEBA | 54.9 | 91.3 | 76.9 | 90.0 | 86.8 | 78.4 | 87.1 | 87.1 | 79.3 | 81.3 |
| GIFT | 56.3 | 90.9 | 78.2 | 91.7 | 88.3 | 79.8 | 88.7 | 88.6 | 81.0 | 82.6 |
| GMM | 59.5 | 93.7 | 81.6 | 94.8 | 91.2 | 83.1 | 91.4 | 91.6 | 84.2 | 85.7 |
| Ours | **61.6** | **95.8** | **83.4** | **96.6** | **93.4** | **85.3** | **93.3** | **92.5** | **86.4** | **87.6** |

*Table 2.* Comparison with state-of-the-art methods on cross-domain class-incremental learning (Order-II).

| Method | Food | SUN397 | Caltech101 | Cars | Imagenet-R | Flowers | OxfordPet | DTD | Aircraft | *Average* |
|---|---|---|---|---|---|---|---|---|---|---|
| Continual-FT | 73.1 | 66.8 | 72.5 | 71.3 | 60.1 | 77.3 | 66.1 | 56.7 | 29.4 | 63.7 |
| LwF | 77.6 | 69.0 | 77.9 | 73.0 | 65.6 | 78.5 | 70.8 | 57.8 | 33.9 | 67.1 |
| iCaRL | 80.6 | 73.2 | 81.1 | 77.4 | 68.8 | 81.9 | 74.5 | 60.7 | 38.1 | 70.7 |
| WiSE-FT | 83.8 | 76.5 | 83.9 | 81.4 | 73.5 | 85.6 | 78.8 | 65.5 | 43.1 | 74.6 |
| ZSCL | 89.6 | 82.8 | 88.3 | 87.5 | 79.2 | 89.1 | 83.0 | 71.2 | 49.5 | 80.1 |
| MoE-Adapters | 90.3 | 83.5 | 89.6 | 88.8 | 80.5 | 89.7 | 84.0 | 72.3 | 50.1 | 81.0 |
| LEBA | 91.0 | 83.9 | 90.1 | 89.4 | 80.7 | 90.2 | 84.8 | 72.8 | 50.5 | 81.5 |
| GIFT | 91.8 | 85.0 | 90.7 | 90.3 | 81.7 | 90.8 | 85.4 | 73.3 | 51.4 | 82.3 |
| GMM | 93.3 | 86.0 | 92.8 | 91.6 | 84.4 | 92.6 | 88.3 | 74.7 | 52.6 | 84.1 |
| Ours | **94.8** | **86.9** | **94.1** | **93.5** | **85.9** | **93.8** | **89.8** | **76.5** | **54.5** | **85.3** |

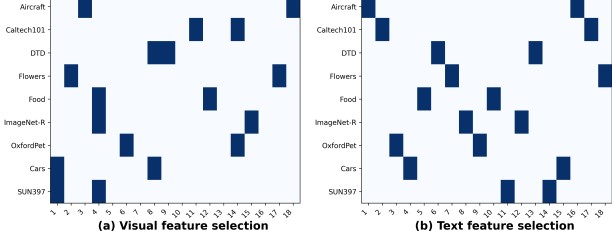

*Figure 1.* Visual expert selection. *Figure 2.* Text expert selection.

additional analysis available in the supplementary material.

**Effectiveness of different modules:** We conduct ablation studies to evaluate the effectiveness of the two core components in PMDO—DKD and STR—with detailed results reported in Table 3. The results show that introducing DKD consistently improves over the baseline across domains, highlighting the value of disentangling what to update under domain shifts. By selectively activating a domain-relevant subset of shared parameters, DKD prevents unnecessary updates to the shared space and leads to more stable adaptation across heterogeneous domains. Building upon DKD, further incorporating STR yields additional and more pronounced gains across datasets. This indicates that, beyond restricting the update scope, it is also important to regulate how to update at domain boundaries, where abrupt distribution shifts can induce directionally inconsistent gradients and unstable optimization. STR constrains update directions and guides optimization along more transferable trajectories, thereby mitigating cross-domain interference.

**Visualization of experts:** We visualize the expert-selection patterns induced by the stability-aware trajectory regulator (STR) for visual and text features in Fig. 1 and Fig. 2. The selection frequencies exhibit a sparse yet diverse behavior: STR activates a small subset of experts while allowing the dominant experts to vary across domains and modalities. This indicates that STR performs multi-directional updates by routing gradients through different expert subspaces, instead of forcing all domains to share a single update direction in the shared parameter space. Such directional diversity is particularly important at domain boundaries, where gradients are more likely to be inconsistent; by distributing updates across multiple expert directions and emphasizing the most transferable ones, STR reduces cross-domain interference and stabilizes optimization.

**Activation rate $\rho$ analysis:** We analyze the effect of the activation ratio $\rho$, which controls the fraction of neurons activated by the soft-mask, as summarized in Table 4. The results reveal a clear non-monotonic trend: moderate activation leads to the most favorable performance, whereas both overly sparse and overly dense masks are suboptimal. When $\rho$ is too small, the mask becomes overly selective and may suppress task-relevant signals, limiting the model's ability to adapt to domain-specific variations. In contrast, when $\rho$ is too large, the mask loses its discriminative gating role and tends to activate many domain-irrelevant channels, weakening the intended separation between useful and noisy updates and making optimization less focused. These findings suggest that $\rho$ serves as a capacity–selectivity knob for soft masking: a moderate value provides sufficient flexibility for adaptation while maintaining structured sparsity to

*Table 3.* Performance comparison of DKD and STR modules in PMDO.

| Method | Aircraft | Caltech101 | DTD | Flowers | Food | ImageNet-R | OxfordPet | Cars | SUN397 | *Average* |
|---|---|---|---|---|---|---|---|---|---|---|
| Baseline | 58.2 | 92.1 | 80.2 | 93.0 | 89.5 | 81.8 | 90.2 | 88.7 | 82.8 | 84.1 |
| +DKD | 59.8 | 93.0 | 81.2 | 94.9 | 90.4 | 82.8 | 92.2 | 89.7 | 84.1 | 85.3 |
| +STR | 58.9 | 92.3 | 80.7 | 94.5 | 90.8 | 83.3 | 91.8 | 90.1 | 84.8 | 85.1 |
| +DKD+STR | **61.6** | **95.8** | **83.4** | **96.6** | **93.4** | **85.3** | **93.3** | **92.5** | **86.4** | **87.6** |

*Table 4.* Performance comparison across different $\rho$ values (%).

| $\rho$ | Aircraft | Caltech101 | DTD | Flowers | Food | ImageNet-R | OxfordPet | Cars | SUN397 | *Average* |
|---|---|---|---|---|---|---|---|---|---|---|
| $\rho$=0% | 58.4 | 93.0 | 80.3 | 93.7 | 90.0 | 82.6 | 90.3 | 89.2 | 83.5 | 84.5 |
| $\rho$=20% | 61.6 | 95.8 | 83.4 | 96.6 | 93.4 | 85.3 | 93.3 | 92.5 | 86.4 | 87.6 |
| $\rho$=50% | 60.2 | 94.2 | 82.1 | 94.9 | 91.9 | 83.7 | 91.9 | 91.0 | 85.2 | 86.1 |
| $\rho$=100% | 59.2 | 93.6 | 81.1 | 94.0 | 91.0 | 83.5 | 90.9 | 90.4 | 84.1 | 85.3 |

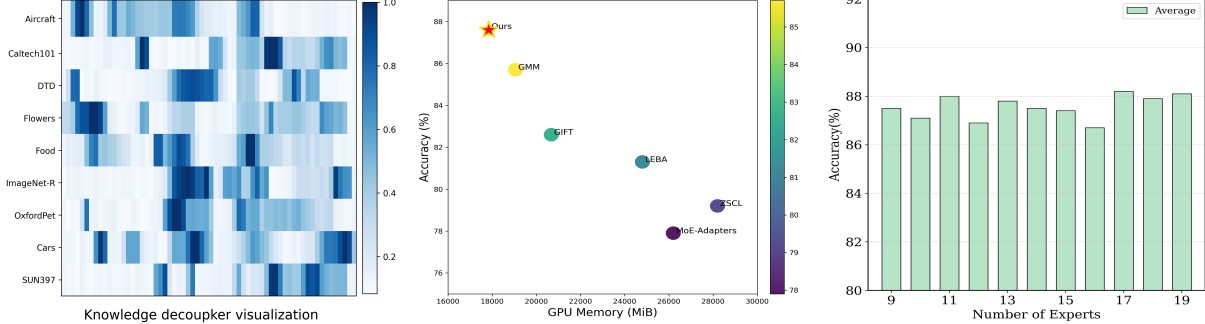

*Figure 3.* The knowledge decoupler visual-*Figure 4.* The ablation experiments of com-*Figure 5.* The ablation experiments of expert number.
ization.                    putational cost.

keep updates targeted and stable across domain shifts.

**Decoupler visualization:** We visualize the learned knowledge decoupler to better understand how the model separates different domains and to support our key contribution of domain-disentangled gating, as shown in Fig. 3. The visualization shows clearly distinguishable activation patterns across datasets: each domain exhibits structured block-like responses, and the high-activation regions shift to different channel locations from one dataset to another. This indicates that the decoupler serves as an explicit mechanism for separating domains via adaptive channel selection, validating its role as a core contributor to robust cross-domain adaptation.

**Computational cost:** To evaluate the computational efficiency of PMDO, we compare the overall performance and GPU memory consumption across representative baselines, as shown in Fig. 4. Compared to prior approaches, PMDO achieves stronger results while requiring less memory overhead, indicating a more efficient use of computational resources. This advantage stems from its selective update and lightweight gating design, which avoids unnecessary full-parameter adaptation and keeps the optimization footprint compact. Overall, the results demonstrate that PMDO offers a favorable trade-off between effectiveness and efficiency, making it well-suited for cross-domain class-incremental learning under practical resource constraints.

**Expert Number $K$ analysis:** We analyze the effect of

the expert number in STR. As shown in Fig. 5, STR exhibits stable behavior across a wide range of expert counts, suggesting that the trajectory regulation mechanism is not overly sensitive to this hyperparameter. When the number of experts is small, the available update directions are limited, which can constrain STR's ability to represent diverse, domain-specific optimization trajectories. Increasing the expert pool enlarges the set of candidate directions/subspaces, allowing STR to perform more fine-grained routing and thus better distribute updates across multiple directions to alleviate cross-domain interference. However, the benefit tends to saturate in the moderate-to-large regime, indicating diminishing returns once the directional capacity becomes sufficient; further increasing the number of experts mainly adds routing and computation overhead and may weaken per-expert specialization due to reduced training signal per expert. Based on this trade-off, we use a moderate expert number in all experiments.

## 5. Conclusion

In this work, we study cross-domain class-incremental learning, where a model can continuously learn new classes across domains while retaining previously learned knowledge in a shared parameter space. We identify a key bottleneck: existing methods often entangle what to update with how to update under domain shifts, which leads to unstable adaptation and severe forgetting. To address this, we propose Parameter-Masked Decoupled Optimization

(PMDO) framework that explicitly separates update scope from update trajectory. PMDO introduces a domain-aware knowledge decoupler to selectively activate domain-relevant shared parameters, avoiding indiscriminate updates and preserving prior representations. It further incorporates a stability-aware trajectory regulation module to steer optimization along transferable and stable directions, suppressing cross-domain interference at domain boundaries. Extensive experiments on multiple benchmarks demonstrate that PMDO consistently outperforms representative baselines and recent state-of-the-art methods, achieving stronger overall performance while maintaining long-term learnability.

## Acknowledgement

This work was supported by the National Natural Science Foundation of China (Grant Nos. 62372238 and 62476133) and the National Science and Technology Major Project of China (Grant No. 2024ZD0524600) and the Fundamental Research Funds for the Central Universities (Grant No. 11300-312200502507).

## Impact Statement

"This paper presents work whose goal is to advance the field of Machine Learning. There are many potential societal consequences of our work, none which we feel must be specifically highlighted here."

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

# A. Derivation of Domain-Aware Knowledge Decoupler

## A.1. Problem Setup

At incremental session $t$, given an image–caption pair $(x_i^t, c_i^t)$, frozen encoders extract cross-modal features as:

$$\mathbf{v}_i^t = \text{Enc}_{\text{img}}(x_i^t), \qquad \mathbf{z}_i^t = \text{Enc}_{\text{txt}}(c_i^t). \tag{14}$$

Concatenating them yields a domain-context representation $[\mathbf{v}_i^t; \mathbf{z}_i^t]$, based on which we infer a layer-wise relevance score via a lightweight network as:

$$\eta^{t,l} = \varepsilon([\mathbf{v}_i^t; \mathbf{z}_i^t]; \theta_p), \qquad \eta^{t,l} \in \mathbb{R}^{c_l}, \tag{15}$$

where $c_l$ denotes the number of neurons (channels/hidden units) at layer $l$.

## A.2. Soft Mask Construction

To make the update scope explicit and learnable across sessions, we maintain a persistent structural mask parameter as:

$$\tilde{\mathbf{M}}^{(l)} \in \mathbb{R}^{c_l}, \qquad \mathbf{M}^{(l)} = \sigma(\tilde{\mathbf{M}}^{(l)}) \in (0,1)^{c_l}, \tag{16}$$

where $\sigma(\cdot)$ is the sigmoid function and $\mathbf{M}^{(l)}$ encodes a session-accumulated update preference.

We fuse the sample-conditioned relevance $\eta^{t,l}$ and the structural prior $\mathbf{M}^{(l)}$ to form a sample-specific update mask:

$$\mathbf{R}^{(l)}(x_i^t) = \underbrace{\text{Top-}k(\eta^{t,l})}_{k\text{-sparse selection}} \odot \underbrace{\mathbf{M}^{(l)}}_{\text{structural prior}}, \qquad \mathbf{R}^{(l)}(x_i^t) \in [0,1]^{c_l}. \tag{17}$$

Here, $\text{Top-}k(\cdot)$ produces a hard $k$-sparse indicator vector (with $k$ active entries); for stable optimization, we optionally employ a differentiable relaxation during training.

## A.3. Masked Optimization and Gradient Gating

Let $\zeta_{\text{task}}$ be the training objective at session $t$ (e.g., classification loss). DKD enforces neuron-wise selective updates by gating gradient flow with $\mathbf{R}^{(l)}$. A convenient formulation is to re-parameterize the effective trainable weight as:

$$\bar{\mathbf{W}}^{(l)} = \mathbf{R}^{(l)}(x_i^t) \odot \mathbf{W}^{(l)}, \tag{18}$$

and compute the forward pass using $\bar{\mathbf{W}}^{(l)}$. Then the gradient w.r.t. the original parameters becomes:

$$\frac{\partial \zeta_{\text{task}}}{\partial \mathbf{W}^{(l)}} = \mathbf{R}^{(l)}(x_i^t) \odot \frac{\partial \zeta_{\text{task}}}{\partial \bar{\mathbf{W}}^{(l)}}. \tag{19}$$

Therefore, parameters associated with $\mathbf{R}_j^{(l)}(x_i^t) \approx 0$ receive negligible gradients and remain effectively unchanged, while those selected by $\mathbf{R}_j^{(l)}(x_i^t) \approx 1$ are allowed to adapt. This yields selective plasticity in the shared parameter space and reduces unnecessary perturbations under domain shifts.

**Remark on Top-$k$.** Since $\text{Top-}k(\cdot)$ is non-differentiable, we adopt a standard straight-through strategy: the hard mask is used in the forward pass, while gradients are back-propagated through the pre-selection scores $\mathbf{s}^{t,l}$. Concretely, we treat $\mathbf{R}^{(l)}$ as a constant in the backward pass and propagate gradients to $\theta_p$ and $\tilde{\mathbf{M}}^{(l)}$ via $\mathbf{s}^{t,l}$.

## A.4. Budget Regularizer and Its Gradient

Overly conservative gating may drive masks toward zero and harm adaptation. To prevent this degenerate behavior, we introduce a budget regularizer that aligns the mean activation ratio with a target budget $\rho$:

$$\zeta_{\text{DKD}}^{(l)} = \Big( \frac{1}{c_l} \sum_{j=1}^{c_l} \mathbf{R}_j^{(l)}(x_i^t) - \rho \Big)^2. \tag{20}$$

This regularizer softly encourages selecting approximately a $\rho$ fraction of neurons at layer $l$.

The gradient of $\zeta_{\text{DKD}}^{(l)}$ w.r.t. the mask entries is:

$$\frac{\partial \zeta_{\text{DKD}}^{(l)}}{\partial \mathbf{R}_j^{(l)}} = \frac{2}{c_l} \left( \frac{1}{c_l} \sum_{q=1}^{c_l} \mathbf{R}_q^{(l)} - \rho \right). \tag{21}$$

Thus, if the average activation is below $\rho$, the term provides a positive push that increases mask activations; otherwise it decreases them.

Further, since $\mathbf{s}^{t,l} = \eta^{t,l} \odot \mathbf{M}^{(l)}$ and $\mathbf{M}^{(l)} = \sigma(\tilde{\mathbf{M}}^{(l)})$, the gradients to the structural mask parameters follow the chain rule:

$$\frac{\partial \zeta_{\text{DKD}}^{(l)}}{\partial \tilde{\mathbf{M}}^{(l)}} = \frac{\partial \zeta_{\text{DKD}}^{(l)}}{\partial \mathbf{R}^{(l)}} \cdot \frac{\partial \mathbf{R}^{(l)}}{\partial \mathbf{s}^{t,l}} \cdot \frac{\partial \mathbf{s}^{t,l}}{\partial \mathbf{M}^{(l)}} \cdot \frac{\partial \mathbf{M}^{(l)}}{\partial \tilde{\mathbf{M}}^{(l)}}. \tag{22}$$

Under the straight-through treatment of Top-$k$, $\frac{\partial \mathbf{R}^{(l)}}{\partial \mathbf{s}^{t,l}}$ is approximated by an identity (or a sparse selector), enabling stable updates to $\theta_p$ and $\tilde{\mathbf{M}}^{(l)}$.

### A.5. Overall Training Objective

We jointly optimize the incremental model parameters and DKD parameters via as:

$$\arg \min_{\Theta, \Gamma} \quad \zeta_{\text{task}} + \lambda \sum_l \zeta_{\text{DKD}}^{(l)}, \tag{23}$$

where $\lambda$ controls the strength of the budget constraint. Intuitively, DKD realizes a principled "what-to-update" mechanism by learning a domain-conditioned subspace (through $\eta^{t,l}$) and a persistent structural prior (through $\tilde{\mathbf{M}}^{(l)}$), while explicitly maintaining sufficient update capacity via the budget regularizer.

## B. Derivation of the Stability-Aware Trajectory Regulator

### B.1. LoRA adaptation and the need for directional control

Consider an incremental weight matrix $W \in \mathbb{R}^{d_v \times d_h}$ with pretrained initialization $W_0$. STR adopts LoRA-style low-rank adaptation:

$$W = W_0 + \Delta W, \qquad \Delta W = AB^\top, \tag{24}$$

where $A \in \mathbb{R}^{d_v \times r}$, $B \in \mathbb{R}^{d_h \times r}$, and $r \ll \min(d_v, d_h)$. Let $\zeta_{\text{total}}$ denote the incremental training objective. The unconstrained gradients of LoRA factors are $\nabla_A \zeta_{\text{total}}$ and $\nabla_B \zeta_{\text{total}}$. Although DKD restricts *which* parameters are updated, under domain shifts the gradients can still be directionally inconsistent, causing unstable optimization even within the selected subset. STR therefore aims to regulate *how* LoRA factors are updated by projecting gradients onto sample-adaptive directional subspaces.

### B.2. Directional subspace experts and projectors

We construct a bank of $K$ direction subspace experts. For expert $k$, we define two orthonormal bases:

$$U_k \in \mathbb{R}^{d_v \times m}, \qquad V_k \in \mathbb{R}^{d_h \times m}, \qquad m \ll d_v, d_h, \tag{25}$$

with $U_k^\top U_k = I_m$ and $V_k^\top V_k = I_m$. These bases induce two orthogonal projectors:

$$P_k^{\text{in}} = U_k U_k^\top \in \mathbb{R}^{d_v \times d_v}, \qquad P_k^{\text{out}} = V_k V_k^\top \in \mathbb{R}^{d_h \times d_h}. \tag{26}$$

Intuitively, $P_k^{\text{in}}$ constrains admissible update directions for $A$ (input-side), while $P_k^{\text{out}}$ constrains those for $B$ (output-side).

### B.3. Sample-conditioned routing and soft expert mixture

Given an input sample $(x_i^t, c_i^t)$ at session $t$, we extract cross-modal features:

$$\mathbf{v}_i^t = \text{Enc}_{\text{img}}(x_i^t), \qquad \mathbf{z}_i^t = \text{Enc}_{\text{txt}}(c_i^t). \tag{27}$$

A lightweight router first maps features into a routing representation via two projection heads:

$$\mathbf{s}_i^v = f_v(\mathbf{v}_i^t; \theta_v), \qquad \mathbf{s}_i^t = f_t(\mathbf{z}_i^t; \theta_t), \tag{28}$$

and outputs expert weights through a softmax over $K$ experts:

$$\mathbf{q}_i^t = \mathrm{softmax}\big(\mathbf{s}_i^v; \mathbf{s}_i^t\big) \in \mathbb{R}^K, \qquad \sum_{k=1}^{K} \mathbf{q}_{i,k}^t = 1, \tag{29}$$

where $g(\cdot)$ is a lightweight fusion MLP (or linear layer) producing $K$ logits. Instead of selecting discrete experts, we form sample-specific *soft* projectors by convex mixing:

$$\bar{P}_i^{\mathrm{in}} = \sum_{k=1}^{K} \mathbf{q}_{i,k}^t P_k^{\mathrm{in}}, \qquad \bar{P}_i^{\mathrm{out}} = \sum_{k=1}^{K} \mathbf{q}_{i,k}^t P_k^{\mathrm{out}}. \tag{30}$$

Note that each $P_k$ is symmetric positive semidefinite (PSD); hence $\bar{P}_i^{\mathrm{in}}$ and $\bar{P}_i^{\mathrm{out}}$ are also PSD. They act as sample-adaptive directional filters on gradients.

## B.4. Projected gradients via subspace-constrained updates

Let $\zeta_{total}$ denote the total training objective, and let $G_A = \nabla_A \zeta_{total}$ and $G_B = \nabla_B \zeta_{total}$ be the unconstrained gradients for the LoRA factors $A$ and $B$. STR restricts updates to routed subspaces by projecting these gradients:

$$\widetilde{G}_A = P^{\mathrm{in}} G_A, \qquad \widetilde{G}_B = P^{\mathrm{out}} G_B, \tag{31}$$

where $P^{\mathrm{in}}$ and $P^{\mathrm{out}}$ are (learned or routed) projection matrices that specify the allowed update directions for $A$ and $B$, respectively. Therefore, STR keeps only the gradient components aligned with the routed subspace and filters out off-subspace components, which helps suppress domain-shifted gradient directions that may interfere with previously learned representations.

## B.5. Entropy-based sharpness regularizer for expert concentration

Although soft mixing improves flexibility, the routing distribution can become overly diffuse at the sample level. We therefore encourage each sample to concentrate probability mass on a small subset of experts by minimizing entropy:

$$\zeta_{\mathrm{STR}} = \frac{1}{N} \sum_{i=1}^{N} H(\mathbf{q}_i^t) = -\frac{1}{N} \sum_{i=1}^{N} \sum_{k=1}^{K} \mathbf{q}_{i,k}^t \log\big(\mathbf{q}_{i,k}^t + \epsilon\big), \tag{32}$$

where $N$ is the batch size and $\epsilon$ is a small constant for numerical stability. The gradient w.r.t. routing weights is:

$$\frac{\partial \zeta_{\mathrm{STR}}}{\partial \mathbf{q}_{i,k}^t} = -\frac{1}{N} \Big( \log(\mathbf{q}_{i,k}^t + \epsilon) + 1 \Big), \tag{33}$$

which pushes large probabilities to become larger and small probabilities to become smaller, leading to sharper, more interpretable expert selections.

## B.6. Overall objective for STR

STR is optimized jointly with the incremental learner by augmenting the training objective:

$$\arg\min_{\Theta, \pi} \ \zeta_{\mathrm{task}} + \alpha \zeta_{\mathrm{STR}}, \tag{34}$$

where $\alpha$ controls the regularization strength. Overall, STR regulates "how-to update" by (i) routing each sample to a mixture of direction subspaces, (ii) projecting LoRA gradients onto the routed subspaces to suppress cross-domain interference, and (iii) sharpening routing distributions to promote multi-directional yet sparse updates.

*Table 5.* Sensitivity to balance factor $\gamma$.

|  | Aircraft | Caltech101 | DTD | Flowers | Food | ImageNet-R | OxfordPet | Cars | SUN397 | *Average* |
|---|---|---|---|---|---|---|---|---|---|---|
| $\gamma=0.1$ | 60.6 | 94.8 | 82.4 | 96.0 | 91.9 | 85.0 | 92.3 | 90.9 | 85.8 | 86.7 |
| $\gamma=0.5$ | 61.6 | 95.8 | 83.4 | 96.6 | 93.4 | 85.3 | 93.3 | 92.5 | 86.4 | 87.6 |
| $\gamma=1$ | 60.5 | 93.8 | 82.6 | 94.5 | 92.3 | 83.4 | 92.2 | 90.6 | 85.6 | 86.2 |

*Table 6.* Sensitivity to balance factor $\alpha$.

|  | Aircraft | Caltech101 | DTD | Flowers | Food | ImageNet-R | OxfordPet | Cars | SUN397 | *Average* |
|---|---|---|---|---|---|---|---|---|---|---|
| $\alpha=0.05\%$ | 60.5 | 94.5 | 82.1 | 95.2 | 92.0 | 84.2 | 92.0 | 91.2 | 85.2 | 86.4 |
| $\alpha=0.1\%$ | 61.6 | 95.8 | 83.4 | 96.6 | 93.4 | 85.3 | 93.3 | 92.5 | 86.4 | 87.6 |
| $\alpha=0.2\%$ | 60.7 | 93.9 | 82.4 | 94.6 | 92.2 | 83.4 | 92.2 | 90.7 | 85.5 | 86.3 |

*Table 7.* Comparison with state-of-the-art methods on cross-domain class-incremental learning (Order-III).

| Method | Cars | Food | OxfordPet | SUN397 | DTD | Aircraft | Flowers | ImageNet-R | Caltech101 | *Average* |
|---|---|---|---|---|---|---|---|---|---|---|
| Continual-FT | 76.7 | 76.6 | 78.4 | 71.9 | 65.4 | 42.5 | 79.3 | 71.2 | 78.4 | 71.1 |
| LwF | 79.1 | 78.3 | 80.8 | 73.2 | 68.2 | 43.8 | 82.1 | 73.2 | 79.9 | 73.1 |
| iCaRL | 81.4 | 80.4 | 82.9 | 74.7 | 70.3 | 45.5 | 84.9 | 75.0 | 81.6 | 75.2 |
| WiSE-FT | 82.5 | 81.1 | 83.9 | 75.4 | 71.5 | 46.4 | 85.5 | 76.7 | 83.4 | 76.4 |
| ZSCL | 85.9 | 83.9 | 86.7 | 78.4 | 75.0 | 49.0 | 88.9 | 78.9 | 86.8 | 79.3 |
| MoE-Adapters | 87.0 | 84.7 | 88.0 | 79.3 | 76.2 | 49.7 | 90.3 | 79.9 | 87.4 | 80.2 |
| LEBA | 88.8 | 86.8 | 90.4 | 81.3 | 77.6 | 52.3 | 92.6 | 81.3 | 89.6 | 82.4 |
| GIFT | 90.1 | 87.7 | 91.3 | 81.8 | 78.9 | 53.6 | 92.9 | 83.0 | 90.5 | 83.2 |
| GMM | 91.2 | 88.5 | 92.4 | 82.9 | 80.3 | 53.9 | 94.2 | 84.0 | 91.4 | 84.3 |
| Ours | **92.3** | **89.4** | **93.6** | **83.7** | **81.3** | **55.2** | **94.9** | **85.1** | **92.2** | **85.3**(+1.0) |

## C. Baselines

We compare PMDO with a diverse set of representative baselines covering conventional class-incremental learning, parameter-efficient adaptation, and recent CLIP-based CD-CIL methods, including Continual-FT, LwF (Li & Hoiem, 2017), iCaRL (Rebuffi et al., 2017), WiSE-FT (Ding et al., 2022), ZSCL (Zheng et al., 2023), and MoE-Adapters (Yu et al., 2024), as well as state-of-the-art approaches such as LEBA (Gu et al., 2026), GIFT (Wu et al., 2025), and GMM (Cao et al., 2024). Specifically, Continual-FT performs naive sequential fine-tuning without explicit anti-forgetting mechanisms; LwF regularizes updates via knowledge distillation from the previous model; iCaRL combines exemplar rehearsal with representation learning; WiSE-FT improves robustness by interpolating weights between pretrained and fine-tuned models; ZSCL leverages CLIP's zero-shot generalization to alleviate knowledge degradation; MoE-Adapters introduces mixture-of-experts adapter routing for domain-specific adaptation; LEBA models relationships among adapters through bridging connections to enhance knowledge reuse; GIFT reduces forgetting via diffusion-synthesized image–text replay and distillation; and GMM employs a generative modeling mechanism to support continual adaptation under domain shifts. We follow official implementations and recommended hyperparameter settings when available to ensure fair comparisons under the same CD-CIL protocol.

## D. Other Results

**Balance factor analysis:** We further study the sensitivity of PMDO to the balance factors $\gamma$ and $\alpha$, as shown in Tables 5 and 6. Overall, PMDO exhibits stable behavior across the tested ranges, indicating that the proposed training objective is not overly sensitive to these hyperparameters. For $\gamma$, an intermediate setting yields the most consistent performance, whereas overly small or overly large values tend to be less favorable. This suggests that $\gamma$ should provide a proper balance between encouraging the desired regularization effect and preserving sufficient flexibility for domain adaptation. Similarly, for $\alpha$, a moderate value achieves the best overall trade-off, while either too weak or too strong weighting can slightly degrade performance, likely due to under-regularization or excessive constraint. Based on these observations, we adopt the intermediate settings for $\gamma$ and $\alpha$ as the default configuration in all experiments.

**Other domain order:** Table 7 reports the comparison on CD-CIL under Order-III, where the domain sequence is rearranged to evaluate robustness to task ordering. PMDO achieves the best average performance under this new order and shows consistently strong results across all evaluated domains, indicating a more reliable stability–plasticity trade-off. This suggests that the proposed decoupled design is effective in reducing cross-domain interference: by separating the selection of update scope from the regulation of update trajectories, PMDO can adapt to new domains while better preserving previously acquired knowledge, leading to stable improvements even under challenging order variations.

*Table 8.* Ablation study of the Top-$k$ ratio under Order-I.

| Top-$k$ ratio | 10% | 30% | 50% | 70% | 100% |
|---|---|---|---|---|---|
| Average (%) | 86.4 | 87.6 | 87.1 | 86.6 | 86.2 |

*Table 9.* Ablation studies on DKD and STR components.

(a) Effect of Top-$k$ in DKD

| Variant | Avg (%) |
|---|---|
| Baseline | 84.1 |
| Baseline + DKD without Top-$k$ | 84.6 |
| Baseline + DKD without soft mask | 84.8 |
| Baseline + Full DKD | 85.3 |

(b) Effect of Projection in STR

| Variant | Avg (%) |
|---|---|
| Baseline | 84.1 |
| Baseline + STR without projection | 84.5 |
| Baseline + STR without LoRA+router | 84.7 |
| Baseline + Full STR | 85.1 |

**Effect of the Top-$k$:** We vary the update ratio from 10% to 100% as shown in Table 8, . The performance first improves and then declines as the update scope increases, with the best result achieved at 30%. This indicates that a too small ratio limits current-domain adaptation, while a too large ratio updates excessive neurons and increases cross-domain interference. Therefore, 30% provides a better balance between plasticity and stability, and is adopted as the default Top-$k$ ratio.

**Component Analysis of DKD:** As shown in Table 9(a), removing either the Top-$k$ selection or the soft structural mask leads to a clear performance drop compared with the full DKD variant. This indicates that the Top-$k$ operator is useful for restricting the update scope to the most relevant neurons, while the soft mask provides structural guidance for domain-aware parameter modulation. By combining both components, DKD can perform more selective and effective adaptation under different domain shifts.

**Component Analysis of STR:** As shown in Table 9(b), removing either the projection constraint or the LoRA-based router degrades the overall performance. This suggests that the projection constraint helps regularize the update trajectory, while the routed low-rank design provides flexible domain-specific update directions. Together, these components enable STR to better balance current-domain adaptation and historical knowledge preservation.

