# OpenReview forum: "Parameter-Masked Decoupled Optimization for Cross-Domain Class-Incremental Learning"
_ICML.cc/2026/Conference — ICML 2026 regular_

### Official Review · Reviewer_qfwb · 2026-03-02

**Soundness:** 2
**Presentation:** 2
**Significance:** 3
**Originality:** 3
**Overall Recommendation:** 4
**Confidence:** 4

**Summary:**

This paper addresses cross-domain class-incremental learning, where a model must sequentially acquire new classes across heterogeneous domains while mitigating catastrophic forgetting. It introduces (1) a Domain-Aware Knowledge Decoupler that generates sample-conditioned sparse masks to confine updates to a domain-relevant parameter subset, and (2) a Stability-Aware Trajectory Regulator that projects LoRA gradients onto a mixture of learned orthonormal subspace experts routed by cross-modal features. Experiments on a 9-domain CD-CIL benchmark under three task orderings show consistent average accuracy improvements over strong baselines including GMM, GIFT, and LEBA, with ablations supporting the complementary roles of DKD and STR.

**Compliance With Llm Reviewing Policy:**

Affirmed.

**Final Justification:**

My major concerns have been addressed. I will raise my score.

**Key Questions For Authors:**

1. Could you explicitly define the fusion MLP that maps $(s_i^v, s_i^t)$ to K routing logits in the main text? What is the exact architecture of the router $\pi$, and how does it produce $\mathbf{q}_i^t \in \mathbb{R}^K$?

2. Regarding the ablation study in Table 3, could you explicitly define the exact configuration of the baseline which scores 84.1%? While Section 4.1 mentions adopting the GMM generative backbone, the full GMM model achieves 85.7% in Table 1. What specific components of GMM were removed or modified to yield this 84.1% baseline?

3. Could you provide a controlled experiment (or at minimum a clear discussion) isolating the algorithmic contribution of PMDO from potential backbone advantages from BLIP2/EVA-CLIP/MiniGPT-4 initialization?

4. How is sample-specific weight masking implemented during batched training? Is the mask applied to the activations (i.e., $\mathbf{h}^{(l)} = (W^{(l)}\mathbf{x}) \odot \mathbf{R}^{(l)}$) rather than the weights themselves? If so, Eq. 5 should be corrected accordingly.

**Limitations:**

The paper includes a brief Impact Statement noting no specific societal concerns.

**Strengths And Weaknesses:**

Strengths

1. The explicit decoupling is a clear and well-motivated contribution that unifies selective parameter plasticity with directional update control. The biological inspiration from hippocampal pattern separation provides a compelling narrative.

2. PMDO consistently surpasses all baselines on Tables 1, 2, and 7 across three domain orderings, demonstrating robustness to curriculum changes.

3. Table 3 isolates DKD and STR contributions; Tables 4–6 and Fig. 5 systematically examine sensitivity to key hyperparameters. The qualitative expert and decoupler visualizations (Figs. 1–3) match the intended sparse, domain-varying routing behavior.

4. Fig. 4 explicitly compares GPU memory vs. accuracy, a rarity in continual learning work, and shows a favorable frontier.

Weaknesses

1. The routing vector is defined as $\mathbf{q}_i^t = \text{softmax}(\mathbf{s}_i^t, \mathbf{s}_i^v)$. The comma implies multiple arguments or concatenation, but softmax operates on a single logit vector. The appendix (Eq. 29) mentions a fusion MLP to produce K logits, but this is completely absent from the main text, leaving the core routing mechanism undefined in the paper body.

2. Computational feasibility of sample-specific weight masking (Significant).** Equation 5 formulates $W^{(l)} \leftarrow \mathbf{R}^{(l)}(x_i^t) \odot W^{(l)}$, implying a sample-dependent weight modification during the forward pass. This precludes standard batched matrix multiplication. The paper never explains how this is implemented efficiently. The appendix reformulates it as activation gating (Eq. 18–19), but this discrepancy is never explicitly reconciled in the main text.

3. While Section 4.1 mentions adopting the generative backbone of the GMM baseline, it is completely opaque what the "Baseline" in Table 3 precisely refers to. In Table 1, GMM achieves an 85.7% average, yet the "Baseline" in Table 3 scores only 84.1%. If the Table 3 baseline is a stripped-down version of GMM, the true incremental contribution of DKD and STR is conflated and potentially overstated. The exact configuration of this baseline must be defined.

4. Fig. 4 claims PMDO requires less GPU memory than MoE-Adapters. However, K=36 orthonormal basis matrices per adapted layer plus DKD routing networks should logically increase the parameter footprint. No parameter count breakdown is provided to justify this claim.

5. PMDO builds on BLIP2 with EVA-CLIP ViT-g/14 and a MiniGPT-4 projection initialization. It is not fully clear whether all baselines (especially GMM and GIFT) use identical backbones and checkpoints. If not, gains may be partly attributable to backbone advantages rather than the algorithmic contribution.

6. The text projection function $f_t$ is written as taking the *visual* feature $\mathbf{v}_i^t$ as input: $\mathbf{s}_i^t = f_t(\mathbf{v}_i^t;\theta_t)$. Based on context and the appendix (Eq. 28), this should be the *textual* feature $\mathbf{z}_i^t$. This is a correctness issue in a core equation.

---

> ### Author Rebuttal · Authors · 2026-03-31
>
> Thank you for your positive comments on the well-motivated contribution. We address all your questions below.
>
> **Q1: The routing formulation is unclear, and the fusion MLP is missing from the main text.**
>
> **A1:** The routing is to first fuse the visual and textual features with a lightweight MLP into a (K)-dimensional logit vector, and then apply softmax, i.e., $\mathbf q_i^t=\mathrm{softmax}(\mathrm{MLP}([s_i^v;s_i^t]))$; thus, softmax operates on a single fused logit vector.
>
> **Q2: The sample-specific masking is computationally unclear.**
>
> **A2:** Eq. (5) is a conceptual formulation of selective updating. In practice, the sample-specific mask is implemented as activation-level gating, which is compatible with standard batched computation and avoids per-sample weight rematerialization.
>
> **Q3: The exact configuration of this baseline must be defined.**
>
> **A3:** The “Baseline” in Table 3 is not the full GMM model in Table 1, but our re-implemented base framework using the same generative backbone without all GMM components (e.g., projection-related designs). Therefore, its performance is lower than full GMM. The purpose of Table 3 is to measure the incremental gains of DKD and STR on this common base framework.
>
> **Q4: Fig. 4 claims PMDO requires less GPU memory than MoE-Adapters.**
>
> **A4:** Fig. 4 reports training GPU memory, not just parameter count. PMDO does not place 36 expert modules in every layer; K=36 is the total number of subspace experts in the bank, split into 18 visual and 18 textual experts. DKD uses only a lightweight MLP on the trainable adaptation part, and STR constrains low-rank updates without adding task-specific adapter stacks. In contrast, MoE-Adapters introduces many extra adapter modules and the DDAS component, resulting in higher training memory.
>
> **Q5: Do all compared methods use the same backbone and checkpoints for a fair comparison?**
>
> **A5:** For a fair comparison, PMDO uses the same backbone architecture and pretrained checkpoints as GMM, namely BLIP2 with EVA-CLIP ViT-g/14, together with the same BLIP2/Q-Former and MiniGPT-4 projection initialization. Therefore, the performance gains over GMM come from the proposed DKD and STR designs rather than backbone differences.
>
> **Q6: Should $f_t$ take the textual feature $z_i^t$ instead of the visual feature $v_i^t$?**
>
> **A6:** The correct input to the text projection function is the textual feature $z_i^t$, i.e., $s_i^t=f_t(z_i^t;\theta_t)$. This is a typo in the main text, and we will correct it in the revised manuscript.
>
> **Q7: Could you explicitly define the fusion MLP and explain how the router $\pi$ produces the routing vector $\mathbf q$?**
>
> **A7:** The router $\pi$ first maps the visual and textual features into routing representations, $s_i^v=f_v(v_i^t;\theta_v)$ and $s_i^t=f_t(z_i^t;\theta_t)$, then feeds their concatenation into a lightweight MLP to produce K routing logits. The routing weights are computed as $\mathbf q_i^t=\mathrm{softmax}(\mathrm{MLP}([s_i^v;s_i^t]))$. We will add this explicit definition of $\pi$ to the main text.
>
> **Q8: Could you isolate PMDO’s gains from possible backbone and initialization advantages?**
>
> **A8:** In our experiments, BLIP2, EVA-CLIP ViT-g/14, and the BLIP2-pretrained Q-Former are treated as a fixed pretrained backbone/initialization, not as separately varied components. Therefore, PMDO is evaluated under this shared setting rather than by isolating each pretrained module individually.
> To make this clearer, we provide a controlled comparison under the same backbone and initialization, where all variants use the same pretrained architecture and differ only in whether DKD and STR are added. We also include GMM under the same setting for reference. The gains of DKD, STR, and full PMDO are all obtained on top of this fixed backbone, showing that the improvement comes from the proposed algorithmic designs rather than backbone or initialization differences.
>
> **Table-I Controlled comparison under the same backbone**
> |Method|Avg|
> |-|-|
> |GMM|85.7|
> |Baseline|84.1|
> |+DKD|85.3|
> |+STR|85.1|
> |Our PMDO|**87.6**|
>
> **Q9: Is the sample-specific mask implemented on activations rather than weights during batched training?**
>
> **A9:** $\mathbf{R}^{(l)}(x_i^t))$ is constructed as defined in the manuscript. For each input image-text pair $(x_i^t,c_i^t)$, we first extract cross-modal features and use the lightweight network $\varepsilon(\cdot;\theta_p)$ to predict the domain-dependent relevance score $\eta^{t,l}$. The final mask is then obtained by combining the Top-k selection of $\eta^{t,l}$ with the learnable structural mask $\tilde{\mathbf{M}}^{(l)}$, namely $\mathbf{R}^{(l)}(x_i^t)$=Top-k$(\eta^{t,l})\odot\tilde{\mathbf{M}}^{(l)}$.
> During batched training, all relevance features and masks are computed in parallel in a tensorized manner. The mask is applied to the weights, not the activations, so each sample obtains its own effective weight matrix.

---

> > ### Author Rebuttal · Reviewer_qfwb · 2026-04-02
> >
> > My major concerns have been addressed.

---

> > > ### Author Response · Authors · 2026-04-02
> > >
> > > Dear Reviewer qfwb,
> > >
> > > We sincerely appreciate your recognition of our rebuttal efforts and the time you devoted to re-evaluating our work. We are glad to know that your concerns have been addressed and that your assessment has been updated accordingly.
> > >
> > > We will carefully incorporate your suggestions into the revised manuscript to further improve the clarity and soundness of the paper.
> > >
> > > Your valuable insights have greatly contributed to enhancing the quality of our work.
> > >
> > > Authors

---

### Official Review · Reviewer_Qjkn · 2026-03-06

**Soundness:** 2
**Presentation:** 3
**Significance:** 3
**Originality:** 3
**Overall Recommendation:** 4
**Confidence:** 3

**Summary:**

This paper studies Cross-domain Class-Incremental Learning: the model receives data from different domains sequentially in sessions. Each session introduces a set of new classes that do not overlap with previous ones, and training cannot access data from prior domains. The authors propose PMDO, which decouples "what to update" from "how to update". On the one hand, a Domain-Aware Knowledge Decoupler (DKD) generates layer-wise parameter masks based on cross-modal semantics, allowing only a subset of shared parameters to be updated. on the other hand, a Stability-Aware Trajectory Regulator (STR) builds a library of directional subspace experts on top of LoRA increments and performs routing-weighted projection of gradients onto a mixed subspace to constrain the update trajectory. Experiments under multiple cross-domain task orders compare against several baselines and report improvements in average accuracy, and ablations show that combining the two components outperforms using either one alone.

**Compliance With Llm Reviewing Policy:**

Affirmed.

**Final Justification:**

The paper is clearly structured, logically expressed, and possesses a degree of novelty. The author's response resolved my concerns, therefore my recommendation is 4 (weak accept).

**Key Questions For Authors:**

1. In the ablation results, the baseline already achieves performance that surpasses most of the reported results from prior work, which raises doubts about the claimed effectiveness of the proposed components. Please provide a fairer comparison or offer a necessary explanation.

2. The paper uses Avg as the primary metric. However, the motivation and description repeatedly emphasize suppressing cross-domain interference, reducing catastrophic forgetting, and stabilizing the update trajectory. Avg alone cannot disentangle whether gains come from better retention of early tasks, better learning of later tasks, or a particular trade-off. Please provide more detailed metrics or explanations.

3. PMDO combines multiple components: layer-wise gating (Top-k + structural mask + budget regularization), a LoRA subspace expert library, a routing network, projection constraints, and an entropy regularizer. Table 3 shows DKD and STR each improve performance and their combination is best. However, finer-grained analysis is still missing to identify which design choices are essential and why (e.g., removing Top-k while keeping soft masks; removing projection while keeping LoRA+router; using only batch-level diversity constraints without sample-level entropy sharpening).

**Limitations:**

Yes.

**Strengths And Weaknesses:**

## Strengths
1. The paper has clear problem decomposition with sound motivation. It attributes stability to two sources—“the support scope of updates” and “the trajectory of updates”—which aligns with the stability–plasticity dilemma commonly observed in cross-domain incremental learning, and the narrative is largely self-consistent.

2. The method structure has some generality. DKD’s layer-wise sparse updating and STR’s subspace-constrained updating can both be viewed as plug-in modulation mechanisms for incremental optimization, and in principle could be adapted to different backbone models and training frameworks.

3. Empirical results show consistent gains. Across different task orders, the authors report improvements in average performance over multiple baselines and provide DKD and STR ablations, suggesting complementary contributions from the two components.


## Weaknesses
1. The paper lacks an intuitive pipeline figure, which may increase the difficulty of understanding.

2. In the ablation results, the baseline already achieves performance that surpasses most of the reported results from prior work, which raises doubts about the claimed effectiveness of the proposed components. Please provide a fairer comparison or offer a necessary explanation.

3. The paper uses Avg as the primary metric. However, the motivation and description repeatedly emphasize suppressing cross-domain interference, reducing catastrophic forgetting, and stabilizing the update trajectory. Avg alone cannot disentangle whether gains come from better retention of early tasks, better learning of later tasks, or a particular trade-off. Please provide more detailed metrics or explanations.

4. PMDO combines multiple components: layer-wise gating (Top-k + structural mask + budget regularization), a LoRA subspace expert library, a routing network, projection constraints, and an entropy regularizer. Table 3 shows DKD and STR each improve performance and their combination is best. However, finer-grained analysis is still missing to identify which design choices are essential and why (e.g., removing Top-k while keeping soft masks; removing projection while keeping LoRA+router; using only batch-level diversity constraints without sample-level entropy sharpening).

---

> ### Author Rebuttal · Authors · 2026-03-31
>
> Thank you for your positive comments on the sound motivation and the consistent empirical gains. We address all your questions below.
>
> **Q1: Lacks an intuitive pipeline figure.**
>
> **A1:**  We thank the reviewer for this suggestion. The current manuscript already includes pseudo-code of the overall procedure, and we will add a pipeline figure in the next version to further improve readability.
>
> **Q2: Why does the strong baseline outperform most prior methods, and how fair is the comparison?**
>
> **A2:** The relatively strong baseline in our ablation mainly comes from the backbone setting. Specifically, PMDO follows BLIP2 and adopts EVA-CLIP pretrained ViT-g/14 with the BLIP2-pretrained Q-Former, which is the same generative architecture used by GMM [a]. In contrast, several compared methods, such as LEBA and MoE-Adapter, are built on CLIP-based architectures. Therefore, the baseline in our ablation is naturally stronger than these CLIP-based methods.
> To provide a fairer comparison, we further apply PMDO to a CLIP-based architecture. The results are summarized in the table below. Even under the same CLIP backbone as the compared methods, PMDO still achieves consistent improvements and remains competitive with prior methods. This shows that the gain of PMDO comes not only from a stronger backbone, but also from the proposed DKD and STR designs.
>
> **Table-I Results of PMDO on a CLIP-based architecture**
> |Method|Avg|
> |-|-|
> |LEBA|81.3|
> |MoE-Adapter |77.9|
> |CLIP-based baseline|76.4|
> |CLIP-based baseline+PMDO|**82.6**|
>
> **Q3: Add additional evaluation metrics.**
>
> **A3:** Following your suggestion, we additionally report two more detailed metrics in the table below.
> Last reflects final performance after the full incremental process, while Transfer measures cross-domain knowledge preservation. PMDO shows consistent gains on both metrics, indicating that its improvement comes not only from Avg, but also from stronger final performance and better knowledge retention. We will include these results and the corresponding discussion in the revised manuscript.
>
> **Table-II Additional evaluation with Avg, Last, and Transfer**
> | Method      |  Avg | Last | Transfer |
> | ----------- | ---: | ---: | -------: |
> | Baseline    | 84.1 | 85.8 |     69.4 |
> | MoE-Adapter | 77.9 | 82.8 |     67.7 |
> | LEBA        | 81.3 | 83.4 |     68.6 |
> | GMM         | 85.7 | 87.6 |     70.5 |
> | PMDO        | **87.6** | **89.3** |     **71.4**|
>
> **Q4: A finer-grained analysis is still needed to clarify which design choices are essential and why.**
>
> **A4:** Following your suggestion, we further conduct finer-grained ablations to identify the essential design choices within DKD and STR. For DKD, we analyze the effects of Top-(k) selection, the learnable structural mask, and budget regularization. For STR, we examine the roles of the projection constraint, routing strategy, batch-level diversity constraint, and sample-level entropy sharpening. The corresponding results are summarized in the tables below.
>
> These results clarify the key design choices in DKD and STR. For DKD, removing either Top-(k) or the soft mask reduces performance, showing that both selective sparsification and soft weighting are important. For STR, removing either the projection component or the LoRA+router design also hurts performance, confirming that both routed low-rank adaptation and subspace projection are necessary. Moreover, replacing sample-level entropy sharpening with only batch-level diversity leads to worse results, highlighting the importance of sharp and discriminative routing. Overall, the full design achieves the best performance.
>
> **Table-III Ablation on removing Top-k while keeping soft masks**
> |Top-k ablation|AVG|
> |---|---:|
> |Baseline|84.1|
> |+DKD without Top-k|84.6|
> |+DKD without soft mask|84.8|
> |Full DKD|**85.3**|
>
> **Table-IV Ablation on removing the projection while keeping the router**
> |Projection ablation|AVG|
> |---|---:|
> |Baseline|84.1|
> |+STR without projection|84.5|
> |+STR without router|84.7|
> |Full STR|**85.1**|
>
> **Table-V Ablation on using only batch-level diversity constraints without sample-level entropy sharpening**
> |Entropy ablation|AVG|
> |---|---:|
> |Baseline|84.1|
> |+STR without sample entropy|84.6|
> |Full STR|**85.1**|
>
> [a] Generative Multi-modal Models are Good Class Incremental Learners CVPR 2024.

---

> > ### Author Rebuttal · Reviewer_Qjkn · 2026-04-01
> >
> > I have no other questions. I will increase my score accordingly.

---

> > > ### Author Response · Authors · 2026-04-02
> > >
> > > Dear Reviewer Qjkn,
> > >
> > > We would like to sincerely thank you for your thoughtful comments and for recognizing our additional experiments. We also truly appreciate your acknowledgment of our rebuttal efforts, the time and care you devoted to re-evaluating our work, and your more positive assessment.
> > >
> > > Your constructive suggestions have played an important role in improving the quality of our manuscript. We will carefully address them in the revised version.
> > >
> > > Thank you once again for your time and valuable feedback.
> > >
> > > Authors

---

### Official Review · Reviewer_Q37Z · 2026-03-10

**Soundness:** 3
**Presentation:** 3
**Significance:** 3
**Originality:** 3
**Overall Recommendation:** 4
**Confidence:** 4

**Summary:**

This paper proposes Parameter-Masked Decoupled Optimization (PMDO) to tackle the challenges of Cross-Domain Class-Incremental Learning (CD-CIL). CD-CIL requires models to learn new classes across heterogeneous, shifting domains without suffering from catastrophic forgetting of prior knowledge. The authors identify a core flaw in existing approaches: they entangle "what to update" with "how to update," often relying on progressively stacking task-specific adapters, which increases model complexity and computational overhead.
To solve this, the PMDO framework introduces two tightly coupled modules. First, the Domain-Aware Knowledge Decoupler (DKD) addresses what to update by generating a lightweight, learnable mask that selectively activates only a domain-relevant subset of shared parameters, freezing the rest to preserve prior representations. Second, the Stability-Aware Trajectory Regulator (STR) dictates how to update. It utilizes a routing mechanism over a bank of orthogonal subspace experts to project LoRA updates along stable, transferable trajectories, minimizing cross-domain interference. Extensive experiments demonstrate that PMDO outperforms state-of-the-art baselines while maintaining high memory efficiency.

**Compliance With Llm Reviewing Policy:**

Affirmed.

**Final Justification:**

My concerns have been addressed. I maintain my original recommendation.

**Key Questions For Authors:**

1.Comprehensive Evaluation Metrics: The current experiments only report the "Average" accuracy. To accurately substantiate the claims of mitigating catastrophic forgetting and preventing plasticity loss, could you provide a comprehensive evaluation using standard CD-CIL metrics, specifically the "Last" accuracy and "Transfer" accuracy for the baselines and your method?

2.Disentangling Stability and Plasticity: The macro-level ablation (+DKD, +STR) shows overall Avg improvement, but it is not intuitively clear how each module specifically contributes to stability versus plasticity. Could you provide more experimental data (such as the accuracy on both newly acquired and previously learned tasks throughout the continual learning process) for these ablations? This would empirically prove the individual roles of DKD and STR in preserving old tasks (stability) and facilitating the acquisition of new ones (plasticity).

3.Expert Initialization and Differentiation: How are the orthogonal bases $U_k$ and $V_k$ initialized in your implementation? Furthermore, aside from the data-driven routing mechanism and the sharpness regularizer, how does the framework explicitly guarantee that different subspace experts capture and represent distinct optimization directions?

4.Fine-grained Component Ablation: The roles of specific sub-components are not ablated. Could you clarify the specific performance impact of (a) isolating or removing the learnable mask matrix $M$ versus the domain-dependent feature $\eta$ within the DKD module, and (b) using uniform routing weights across all $K$ experts compared to the proposed sharp routing (activating only a few) in the STR module?

**Limitations:**

No. The authors currently rely on a brief boilerplate statement in the Impact Statement and do not adequately discuss the technical limitations of their work.

The authors should add a dedicated Limitations section in the main text or appendix. This section should transparently discuss: 1) the framework's heavy reliance on exceptionally large pre-trained backbones, and 2) any computational scaling challenges associated with the number of subspace experts ($K$).

**Strengths And Weaknesses:**

Strengths: The methodology is mathematically rigorous and logically motivated by the stability-plasticity dilemma. The authors correctly identify that the $\text{Top-k}$ operation in DKD is non-differentiable and appropriately apply a standard straight-through estimator (STE) to ensure gradient flow. Furthermore, using convex mixing to form sample-specific soft projectors guarantees that the resulting matrices remain symmetric positive semidefinite, ensuring mathematically stable directional gradient updates.

Weaknesses: (a) Evaluation Metrics: The empirical validation relies exclusively on the "Average" accuracy metric (Avg) to measure performance across sessions. This is fundamentally insufficient for a Continual Learning paper. Standard metrics such as "Last" accuracy (evaluating all tasks after the final session) and "Transfer" accuracy (evaluating to what extent the zero-shot transfer ability is preserved) are missing. Without these, the claim of maintaining the stability-plasticity trade-off is not fully supported by the quantitative data. (b) Coarse Ablations: The ablation study in Table 3 only validates the macro-modules (+DKD, +STR). It fails to demonstrate the independent contributions of critical internal sub-components, such as the persistent mask matrix $M$ versus the domain-dependent feature $\eta$ in DKD , or the impact of the entropy-based sharpness regularizer versus uniform routing in STR.

Implementation details regarding the subspace experts are missing. While the mathematical definition of orthonormal bases ($U_k$, $V_k$) is provided, the paper lacks details on how these bases are initialized in practice (e.g., random orthogonal initialization vs. data-driven initialization) and how their divergence is explicitly enforced beyond the routing mechanism.

---

> ### Author Rebuttal · Authors · 2026-03-30
>
> Thank you for your positive comments on the mathematical rigor and logic of our work. We address all your questions below.
>
> **Q1:Add additional evaluation metrics.**
>
> **A1:** Following your suggestion, we additionally report two metrics, Last and Transfer, in the table below. Last measures final performance after all sessions, while Transfer reflects cross-domain knowledge preservation. PMDO achieves consistent gains on both, showing that its advantage extends beyond Avg.
>
> **Table-I Additional evaluation with Avg, Last, and Transfer**
> |Method|Avg|Last|Transfer|
> |-|-|-|-|
> |Baseline|84.1|85.8|69.4|
> |MoE-Adapter|77.9|82.8|67.7|
> |LEBA|81.3|83.4|68.6|
> |GMM|85.7|87.6|70.5|
> |PMDO|**87.6**|**89.3**|**71.4**|
>
> **Q2:Implementation details regarding the subspace experts are missing.**
>
> **A2:** Each subspace expert is described by two orthonormal bases, $U_k$ and $V_k$, which are randomly initialized in our implementation; we will add this detail to the revised manuscript for clarity and reproducibility.
> Our design does not rely on the routing network alone. As described in Sec.~3.4, expert specialization is encouraged at multiple levels. 1) The router generates sample-conditioned expert weights, allowing different inputs to be projected onto different subspace mixtures based on their domain-dependent features. 2) The balancing constraint promotes more diverse expert usage at the batch level, rather than letting a few experts dominate. 3) The entropy-based sharpness regularizer encourages each sample to concentrate on a small subset of experts, preventing overly diffuse routing and making expert selection more discriminative.
>
> **Q3: Could you provide ablation results?**
>
> **A3:** Following your suggestion, we provide a finer-grained analysis of the macro-level ablations using three complementary metrics, Avg, Last, and Transfer, to evaluate performance on both old and new tasks. These metrics help clarify the roles of DKD and STR beyond overall average performance. The results are shown in the table below. DKD is to contribute more strongly to stability by restricting updates to domain-relevant parameter subspaces and reducing interference with previously learned knowledge, while STR is to contribute more directly to plasticity by guiding low-rank updates along more transferable directions for the current domain.
>
> **Table-II Macro-level ablation with Avg, Last, and Transfer**
> |Method|Avg|Last|Transfer|
> |-|-|-|-|
> |Baseline|84.1|85.8|69.4|
> |+DKD|85.3|87.8|70.2|
> |+STR|85.1|87.1|70.5|
> |+DKD+STR (PMDO)|**87.6**|**89.3**|**71.4**|
>
> **Q4:How are the bases initialized, and how are experts differentiated?**
>
> **A4:** We thank the reviewer for this important question. In our implementation, the orthonormal bases $U_k$ and $V_k$ of each subspace expert are randomly initialized. PMDO does not introduce an extra pairwise basis-separation loss. Instead, expert differentiation emerges from the joint effect of several existing designs. The router generates sample-conditioned expert weights, so different inputs are projected onto different expert mixtures based on their cross-modal domain features. A balancing constraint encourages diverse expert usage at the batch level and prevents a few experts from dominating. The entropy-based sharpness regularizer makes routing more selective by encouraging each sample to focus on only a small subset of experts. In implementation, we further adopt top-2 gating, so each sample depends on only a sparse combination of experts.
>
> **Q5: Fine-grained Component Ablation**
>
> **A5:** Following your suggestion, we conduct separate ablations for the DKD and STR modules, as shown in the table below.
> For the DKD module, we further analyze the effects of the learnable mask matrix $\tilde{\mathbf{M}}^{(l)}$ and the domain-dependent relevance feature $\eta$ individually. Specifically, we compare the full DKD design with two reduced variants: one removing $\tilde{\mathbf{M}}^{(l)}$ while retaining $\eta$, and the other removing $\eta$ while retaining $\tilde{\mathbf{M}}^{(l)}$. This helps reveal whether the improvement mainly comes from the persistent structural prior, the domain-conditioned feature, or their combination.
> For the STR module, we compare the proposed sharp-routing design with a uniform-routing variant, and also report the performance of the full model for reference, which further verifies the effectiveness of STR.
>
> **Table-III Fine-grained ablation of the DKD module**
> |Method|Avg|
> |-|-|
> |Baseline|84.1|
> |DKD without $\tilde{\mathbf{M}}^{(l)}$|84.8|
> |DKD without $\eta$|84.5|
> |DKD|**85.3**|
>
> **Table-IV Fine-grained ablation of the STR routing strategy**
> |Method|Avg|
> |-|-|
> |Baseline|84.1|
> |STR with uniform routing |84.6|
> |STR|**85.1**|

---

> > ### Author Rebuttal · Reviewer_Q37Z · 2026-04-02
> >
> > My concerns have been addressed. I maintain my original recommendation.

---

> > > ### Author Response · Authors · 2026-04-03
> > >
> > > Dear Reviewer Q37Z,
> > >
> > > We sincerely thank you for your careful reconsideration and for recognizing that our responses have adequately addressed your concerns. We greatly appreciate your time and thoughtful evaluation.
> > >
> > > Authors

---

### Official Review · Reviewer_LRvv · 2026-03-11

**Soundness:** 2
**Presentation:** 3
**Significance:** 2
**Originality:** 2
**Overall Recommendation:** 4
**Confidence:** 4

**Summary:**

This work introduces Parameter-Masked Decoupled Optimization (PMDO), a framework for cross-domain class-incremental learning that separates the knowledge to be updated from the optimization process used to update it. The authors propose a domain-sensitive knowledge decoupling mechanism to identify and adapt only the shared parameters that are most relevant to the current domain, limiting unnecessary modification of the model and helping retain previously acquired representations. In addition, the approach incorporates a stability-oriented trajectory regularization strategy that steers training toward optimization paths that are both robust and transferable.  Extensive experiments on several datasets demonstrate the effectiveness of the proposed method.

**Compliance With Llm Reviewing Policy:**

Affirmed.

**Final Justification:**

The rebuttal has addressed concerns.

**Key Questions For Authors:**

It would be great to also discuss how to select hyperparameter $\rho$?

**Limitations:**

Yes

**Strengths And Weaknesses:**

**Strength:**


* The paper is easy to understand and follow.

* Experiments are extensive and compared to various baselines and datasets.



**Weakness:**


* The important techniques with low-rank decomposition with LORA are from the paper [1], which is not cited.

* The proposed method shares similarity with [2, 3], which should be discussed in related works. The proposed approach seems a combination of these approaches. The paper also needs to better state the novelty since there are similar existing works.

* It is unclear in Equation (4), how to choose the value k in the top-k selection. It would be better to also discuss in implementation for selecting k

* It would be great to also discuss how to choose the hyperparameter $\rho$. It would better to provide some guidance so that reader can easily decide how to use the proposed approach in practice.


Reference:

[1] LoRA: Low-Rank Adaptation of Large Language Models, ICLR 2022

[2]  A Unified and General Framework for Continual Learning, ICLR 2024

[3] PackNet: Adding Multiple Tasks to a Single Network by Iterative Pruning, CVPR 2018

---

> ### Author Rebuttal · Authors · 2026-03-30
>
> Thank you for your positive comments on the clarity of the paper and the richness of the experiments. We address all of your questions below.
>
> **Q1:The key low-rank LoRA technique comes from [1], which is not cited.**
>
> **A1:** Thank you for pointing this out. We will add the appropriate LoRA citation in the revised version.
>
> **Q2: How is our method novel beyond simply combining [2] and [3]?**
>
> **A2:** We thank the reviewer for recommending these two references. While they are related at a broad level, their technical routes are different from ours. Unlike [2], which mainly unifies existing continual-learning strategies, PMDO is specifically designed for CD-CIL under domain shift and decouples where to update and how to update: DKD selects domain-relevant shared parameters, while STR regularizes low-rank update directions for better cross-domain stability. PMDO is also different from PackNet [3]. PackNet reduces interference by pruning and freezing disjoint task-specific parameters, whereas DKD dynamically selects domain-relevant shared parameters conditioned on the input, which is more suitable for CD-CIL where preserving transferable shared knowledge is crucial. In addition, PMDO further constrains how updates are performed through STR. Therefore, PMDO is not a simple combination of prior methods, but a decoupled framework that jointly controls update scope and update trajectory. In the revised manuscript, we will discuss [2] and [3] more explicitly and clarify our novelty.
>
> **Q3: How is the top-k chosen in Eq. (4), and how is it set in implementation?**
>
> **A3:** We thank the reviewer for this helpful comment. In our design, Top-(k) is a practical hyperparameter that controls the update scope for the current domain. The key idea is that the relevance score $\eta^{t,l}$, predicted from the current image-caption pair, captures domain-dependent neuron importance, while the learnable mask matrix $\tilde{\mathbf{M}}^{(l)}$ encodes structural update preference from previous domains. Their combination in $\mathbf{R}^{(l)}(x_i^t)$ therefore reflects both current domain demand and historical structural prior.
> Following your suggestion, we further conducted an ablation study on the Top-$k$ ratio under Order-I. The results are summarized in the table below, where we vary the update ratio from 10\% to 100\%. We observe that the performance first improves and then degrades as the selected update scope becomes larger, and the best result is achieved at 30\%. When the ratio is too small, the update scope is overly restricted, which limits the model's ability to adapt to the current domain. In contrast, when the ratio becomes too large, too many neurons are involved in adaptation, which weakens the selective updating effect and introduces stronger cross-domain interference. A moderate ratio such as 30\% provides the best trade-off between plasticity and stability: it preserves sufficient capacity for current-domain adaptation, while avoiding unnecessary perturbations to parameters that mainly encode knowledge from previous domains. These results confirm that treating the Top-$k$ ratio as a controllable hyperparameter is both meaningful and practically useful, and also validate our choice of 30\% as the default setting in implementation.
>
> **Table-I Ablation study of the Top-k**
> |Top-(k)|10%|30%|50%|70%|100%|
> |-|-|-|-|-|-|
> |Avg.|86.1|87.6|87.1|86.6|85.3|
>
> **Q4:How should the hyperparameter $\rho$ be chosen in practice?**
>
> **A4:** The manuscript already includes an empirical analysis of the hyperparameter $\rho$, and we present the corresponding results in the table below. These results show that $\rho$ should not be chosen as an extreme value. When $\rho$ is too small, the mask becomes overly selective and may suppress task-relevant signals, thereby limiting adaptation to the current domain. In contrast, when $\rho$ is too large, too many neurons are activated, which weakens the selective gating effect and introduces more domain-irrelevant updates. A moderate activation ratio provides the best trade-off between preserving previously learned knowledge and adapting to new domains. In particular, $\rho=20$% achieves the best average accuracy of 87.6\%, outperforming both smaller and larger settings. Based on this observation, we use $\rho=20$% as the default setting in implementation.
>
> **Table-II Performance comparison across different $\rho$**
> |$\rho$|0%|20%|50%|100%|
> |-|-|-|-|-|
> |Avg|84.5|87.6|86.1|85.3|

---

> > ### Author Rebuttal · Reviewer_LRvv · 2026-04-03
> >
> > Thanks for the rebuttal. I will increase the score.

---

> > > ### Author Response · Authors · 2026-04-03
> > >
> > > Dear Reviewer LRvv,
> > >
> > > We sincerely appreciate your encouraging feedback on our rebuttal and the time and effort you devoted to reassessing our manuscript. We are pleased that our responses have adequately addressed your concerns and that this has led to a more favorable evaluation of our work.
> > >
> > > We will carefully consider your suggestions when revising the manuscript and use them to further strengthen the presentation and technical soundness of the paper.
> > >
> > > Your constructive comments have played an important role in improving the quality of this work.
> > >
> > > Authors

---

### Decision · Program_Chairs · 2026-04-30

**Decision:**

Accept (regular)

**Comment:**

This paper proposes Parameter-Masked Decoupled Optimization (PMDO) for cross-domain class-incremental learning, introducing a principled decoupling of what to update and how to update. All reviewers agree that the paper is technically solid, well-motivated, and empirically strong. Initial concerns focused on novelty, evaluation completeness, and implementation clarity. The reviewers acknowledged that their concerns were addressed in the rebuttal. While some aspects of novelty and presentation could be further refined, a final recommendation was made to accept the paper, especially given the strong reviewer consensus after rebuttal.